

# Using observations of surface fracture to address ill-posed ice softness estimation over Pine Island Glacier

Trystan Surawy-Stepney[1], Stephen L. Cornford[2], and Anna E. Hogg[1]

[1]School of Earth and Environment, University of Leeds, United Kingdom
[2]Department of Geographical Sciences, University of Bristol, United Kingdom

**Correspondence:** T. Surawy-Stepney (t.surawystepney@leeds.ac.uk)

**Abstract.** Numerical models used to simulate the evolution of the Antarctic Ice Sheet require the specification of basal boundary conditions on stress and local deviations in the assumed material properties of the ice. In general, scalar fields representing these unknown components of the system are found by solving an inverse problem given observations of model state variables - typically ice flow speed. However, these optimisation problems are ill posed, resulting in degenerate solutions and poor condi-
tioning. In this study, we propose the use of fracture and strain rate data to provide prior information to the inverse problem, in an effort to better constrain the inferred ice softness compared to more heuristic regularisation techniques. We use Pine Island Glacier as a case study and consider both a 'snapshot' inverse problem in which ice softness and basal slip parameters are sought simultaneously over the glacier as a whole, and a 'time-dependent' problem in which ice softness alone is sought over the floating ice shelf at regular intervals. In the first case, we construct a prior encoding the assumption that the ice softness will
be close to our initial guess except from where we see fractures or high shear strain rates in satellite data. We investigate the solutions and conditioning of this data-informed inverse problem versus alternatives. The second proposed method makes the assumption that changes to ice softness occurring on monthly-to-annual timescales will be dominated by the fracturing of ice. We show that these methods can result in softness fields on floating ice that visually mimic fracture patterns without significantly affecting the quality of the solution misfit, perhaps leading to greater confidence in the softness fields as a representation
of the true material properties of the ice shelf.

## 1  Introduction

Large-scale ice sheet models commonly treat ice within the paradigm of continuum mechanics - as a shear thinning viscous fluid; an approach that has been successful in modelling the behaviour of large ice masses relatively cheaply (e.g. Seroussi et al. (2020)). Within this framework, the flow of the ice can be accounted for in large part by a balance between gravity, viscous
stress due to internal deformation and frictional stress at ice/bedrock interfaces. To close the system and allow the model to solve for ice speed, equations relating viscous and frictional stresses to ice speed are specified, informed by laboratory data and physical arguments.





The former 'constitutive relation' very often takes the form of Glen's flow law:

$$\tau_{ij} = 2\eta\dot{\varepsilon}_{ij}, \quad \text{where } \eta = \frac{1}{2}A(T)^{-\frac{1}{n}}\epsilon^{\frac{1}{n}-1} \tag{1}$$

where $\tau_{ij}$ is the deviatoric stress tensor, $\dot{\varepsilon}_{ij}$ is the strain rate tensor, $\epsilon$ is its second invariant, $\eta$ is the strain-rate-dependent effective ice viscosity, $A(T)$ is a temperature-dependent rate factor and the exponent $n \approx 3$. It is possible to treat $A(T)$ and/or $n$ as free parameters that can be fitted to observations, given the uncertainties involved in both and the different physical mechanisms that distinguish them. Here, we consider the approach in which these are prescribed *a priori* and a 'stiffness' field $\phi(x)$, that scales the effective ice viscosity $\eta$, is defined over the domain to account for unknown deviations in the expected ice rheology. Used in this way, $\phi$ approximates the effect of uncertainties in the temperature and thickness fields, regional changes in the temperature dependence of Glen's flow law, deviations from the assumed isotropy of creep deformation and, of particular interest to this study, fractures in the ice at different lengthscales.

The relation between frictional stress and basal sliding speed is known as a 'sliding law', and has a functional form that depends on a number of often poorly constrained factors such as the expected amount of deformation around topographic features in the bed, sliding over smooth bedrock, and shearing of the sub-glacial till. A single sliding law is often combined with a spatially varying 'basal slip' parameter $C(x)$ to approximate this stress:

$$\boldsymbol{\tau}_b = Cf(\boldsymbol{u}). \tag{2}$$

Taken together, the equations considered here take the form of the shallow-stream approximation to the Cauchy momentum equations:

$$\boldsymbol{\nabla} \cdot [\phi h \bar{\eta} (\boldsymbol{\nabla u} + (\boldsymbol{\nabla u})^{\top} + 2(\boldsymbol{\nabla} \cdot \boldsymbol{u})\mathcal{I})] - Cf(\boldsymbol{u}) - \rho_i gh\boldsymbol{\nabla} s = 0, \tag{3}$$

where $\boldsymbol{u} = (u_x, u_y)^{\top}$ is the horizontal velocity, $\bar{\eta}$ is the vertically-integrated effective ice viscosity, $\rho_i$ is the density of ice, $h$ is the ice thickness and $s$ is the ice surface. In this study we use a linear sliding law $f(\boldsymbol{u}) = \boldsymbol{u}$ for ease of computing adjoint sensitivities during the inverse problem. In this article, we also refer to the "softness" field $\varphi$ - related to the stiffness by $\varphi = (1-\phi)$.

In order to simulate real ice masses accurately, the fields $C$ and $\phi$ are inferred simultaneously from observations of ice speed using inverse methods - a suite of techniques for inferring model control parameters from observed state variables (MacAyeal, 1992) - (e.g. Petra et al. (2012); Arthern et al. (2015); Cornford et al. (2015); Gudmundsson et al. (2019)). Unfortunately, this inverse problem is ill-posed: the two fields $(C, \phi)$ are calculated from the single observed field $u$ (the problem is "underdetermined") and the results are highly dependent on noise in the input data (the problem, at least in its discrete form, is "ill-conditioned"). To obtain reliable control fields, it is beneficial to replace this ill-posed problem by a nearby well-posed one before solving it. The problem is sometimes simplified by solving for $C$ only on grounded ice, and $\phi$ on floating ice, thereby separating the two fields spatially removing a portion of the degeneracy that arises from the mixing of $C$ and $\phi$ (e.g. Goldberg



et al. (2019)). However, though you would often expect $C$ to be the dominant control on grounded ice speed, there is little
reason to be especially confident in the guess of $\phi = 1$, and getting this wrong can have consequences for transient simulations. Another approach, and one that shall be taken here, is to regularise the solution by providing additional constraints on the control fields. Such a regularised inverse problem takes the form of the following optimisation:

$$(C, \phi) = \underset{C, \phi}{\operatorname{argmin}} \left\{ \mathcal{J}_m(u, u_o) + \alpha_C \mathcal{J}_C(C) + \alpha_\phi \mathcal{J}_\phi(\phi) \right\}, \quad \text{s.t.} \quad G(u, C, \phi) = 0 \tag{4}$$

where $\mathcal{J}_m(u, u_o) = \|u - u_o\|_2^2$ is a misfit function calculating the distance of the modelled ice speed $u$ from the observed ice speed $u_o$, $\mathcal{J}_C$ and $\mathcal{J}_\phi$ are regularisation functions for the $C$ and $\phi$ fields, with strengths controlled by the parameters $\alpha_C$ and $\alpha_\phi$ respectively, and $G(u, C, \phi) = 0$ are the momentum balance equations (3).

A popular approach, aimed at improving the conditioning of the problem by suppressing the amplification of high-frequency components of the input data, is to use Tikhonov regularisation in a form that favours either low spatial frequency or low amplitude components of the solution (e.g. Morlighem et al. (2013); Habermann et al. (2013); Brinkerhoff and Johnson (2013); Cornford et al. (2015)), e.g.:

$$\alpha_\phi \mathcal{J}_\phi(\phi) = \alpha_\phi \int_\Omega |\nabla \phi|^2 d^2 \boldsymbol{x}. \tag{5}$$

However, this kind of regularisation is entirely heuristic and, when it comes to distinguishing $C$ and $\phi$, relies on assumed differences in the lengthscales over which changes in the control fields can influence strain rates. Generally, in regions without significant shear, these lengthscales are not easily distinguished, and degeneracies between solutions for $C$ and $\phi$ proliferate. Additional difficulties arise when a control field contains distinct contributions with different spatial frequencies. For example, uncertainty in englacial temperature can vary on the scales of long-term atmospheric or geothermal heat sources, or over the width of a shear margin. Often, an imperfect but acceptable lengthscale is found by searching parameter space informed by heuristics such as L-curve analysis (Hansen and O'Leary, 1993; Hansen, 1994).

The aim of this study is to investigate whether the introduction of genuine prior information into the inverse problem can result in substantively different solutions, and whether these solutions are more appealing than those found using other, heuristic regularisation methods.

Previous studies have investigated instances in which softness fields found through solving inverse problems have mirrored observed fracture features (Borstad et al., 2013; Surawy-Stepney et al., 2023a) - suggesting that the presence of fractures has the potential to dominate $\phi$. With recent advancements in observational methods for locating fractures in remote sensing data (Lai et al., 2020; Izeboud and Lhermitte, 2023; Zhao et al., 2022; Surawy-Stepney et al., 2023b), we are moving towards reliable data that can be used to inform us at least about this specific component of the softness field. Ranganathan et al. (2021)



showed previously that the use of strain rate data to weight the regularisation of $C$ and $\phi$ has the potential to reduce mixing
between these control fields. The work presented here follows quite naturally from these results.

Here, we investigate two ways in which fracture and strain-rate observations can be used to inform the inverse problem
to replace or complement existing heuristic methods. The first is to use fracture maps (obtained from Sentinel-1 imagery -

described in Surawy-Stepney et al. (2023b)) along with estimates of surface strain-rates to construct a prior distribution for
$\phi$ for use in snapshot inverse problems (single optimisations carried out for a set of geometry and speed data collected at a
specific instant in time). This prior simply says that we expect $\phi \approx 1$ away from regions of observed fracture, or where there
are high shear strain rates (which can contribute the effects of enhanced anisotropy, shear heating and microfracturing to $\phi$). In
practise, this is equivalent to a form of Tikhonov regularisation using a diagonal Tikhonov matrix with entries weighted away

from where we expect soft ice.
We also investigate the use of timeseries of fracture maps in constraining the solutions to inverse problems carried out over
multiple timesteps. The use of fracture maps as a prior in the snapshot inverse problems makes an assumption about the relative
contributions of different uncertainties to $\phi$. For example, we have to have a certain amount of trust in the 3D temperature field
we use. A more easily justified belief is that *changes* to $\phi$ on monthly-to-annual timescales are dominated by the fracturing

of ice, as other contributions to $\phi$ are likely to vary on significantly longer timsescales. With this in mind, we initialise the
inverse problem with heuristic regularisation, before imposing a regularisation that penalises the changes to $\phi$ except where
we have seen the evolution of fractures in the observational data. We show, with this method, that one can generate softness
fields that mimic, in certain ways, the changing fracture patterns on the Pine Island Ice Shelf between 2016 and 2021, without
substantially affecting the misfit of the problem. This may have potential uses in constraining models that aim to evolve softness

fields in response to englacial stresses.

## 2    Methods

The simulations presented in this article were performed using the BISICLES ice sheet model (Cornford et al., 2013). This is an
adaptive mesh, finite volume model which we choose here to solve discretized versions of the two-dimensional shallow-stream
equations (3). Each simulation is carried out over Pine Island Glacier in the Amundsen Sea Sector of West Antarctica with

a domain encompassing the whole present-day drainage basin (Zwally et al., 2012). This region was chosen as it represents
a potentially strong correspondence between fracturing and ice softness, given the abundant crevasses in the shear margins,
upstream of the grounding line and the regular formation of rifts near the terminus, as well as the established dynamic impact
of some of this fracturing (Joughin et al., 2021; Sun and Gudmundsson, 2023). Across the rest of Antarctica, we expect
the link between the dynamics of ice and the extent of fracturing to be weaker in general. We use a form of the rate factor

$A(T)$ described in Cuffey and Paterson (2010), with an internal energy field generated using a $100\,000\,\mathrm{year}$ calculation
in which surface temperature, thickness and velocity are held at present day values and the combined ice temperature and
moisture fraction field $E = CT + Lw$ evolves toward equilibrium. We used geometry defined by BedMachine-v3 (Morlighem,




2022), with prescribed calving front positions extracted from Sentinel-1 backscatter images. Each simulation used velocity and fracture data from within a five-year period between November 2016 and November 2021. We used 200 m resolution, monthly-averaged ice velocity observations made using feature tracking applied to Sentinel-1 image pairs (Wuite et al., 2021) (https://cryoportal.enveo.at/data/) as the input data to the cost function and to estimate shear strain rates. Crevasse data was generated according to the methods described in Surawy-Stepney et al. (2023b). The inverse problem we consider at each stage takes the form of eq. (4) and is solved in BISICLES using a non-linear conjugate gradient method (Cornford et al., 2015).

## 2.1 Fracture data assimilation in snapshot inverse problems

The snapshot problem we consider is the joint estimation of $C$ and $\phi$ over Pine Island Glacier in May 2019. We use mean velocities over the month and median composite fracture maps.

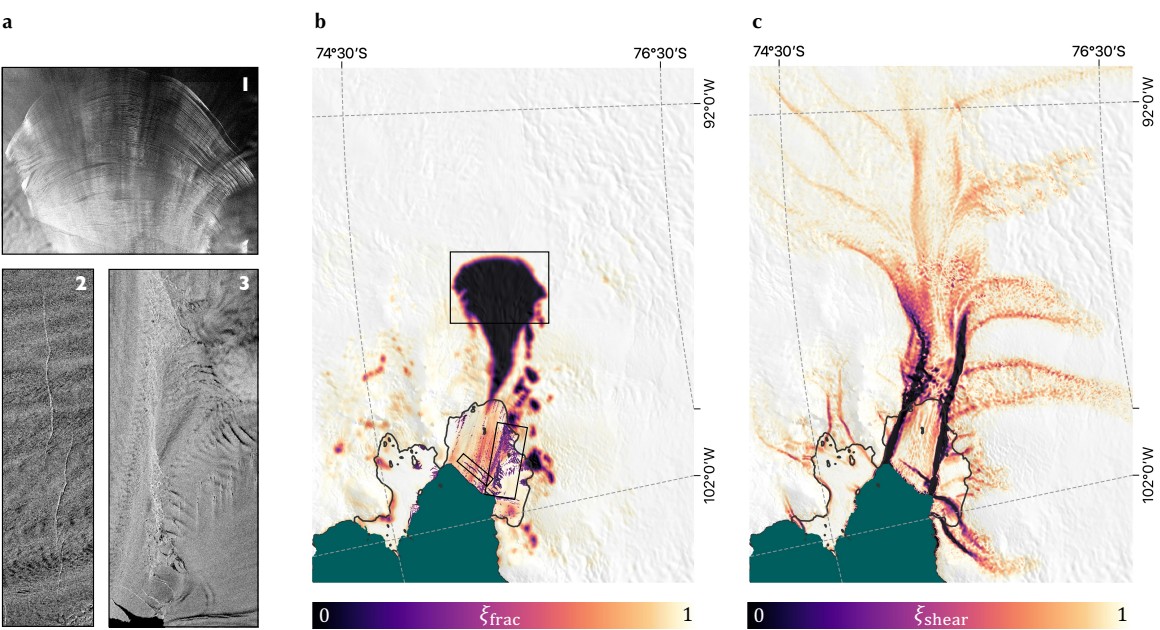

**Figure 1.** Contributions to the field $\xi$, representing, in our prior for the softness field, where we have observations of surface fracture or high shear strain rates. a) SAR backscatter images over grounded and floating parts of Pine Island Glacier from May 2019 showing regions of visible crevassing: 1) surface crevasses on the grounded ice, 2) two almost-connected rifts near the Pine Island calving front, 3) the heavily 'damaged' southern shear margin of Pine Island Ice Shelf. b) The component of $\xi$ due to the observation of crevasse features, made from fracture maps developed in Surawy-Stepney et al. (2023b). Black boxes anticlockwise from the top show the locations of the SAR images a1, a2 and a3 respectively. c) The component of $\xi$ due to the presence of high shear strain rates. Background images to b and c are the MODIS Mosaic of Antarctica (Haran et al., 2021), and grounding lines (shown in black) are according to Rignot et al. (2016).

To construct a prior for $\phi$, we first form a field $\xi$ which is 0 in regions which have high shear strain rates or where fractures have been observed and 1 elsewhere. For the surface fracture contribution to $\xi$, we use monthly mosaics of fracture maps covering grounded and floating ice - slightly smoothed to produce contiguous fracture fields on the grounded ice. We call





these fracture maps $f_i$. The fracture-map contribution to $\xi$ is simply $\xi_{\text{frac}} = 1 - f_i$ (Fig. 1 b). There are a few things to note in these fracture data of potential relevance to the stress-balance of the glacier. Firstly, we see a large contiguous area of surface fractures extending upstream from the grounding line and widening to cover a region in which previous studies have suggested membrane stresses are important in the stress-balance as basal stresses become small (Joughin et al., 2009). SAR images of

this region show uniform coverage by closely-spaced surface fractures, almost identical in appearance (Fig. 1 a1). Additionally, there is a rift (really, two rifts that are almost connected) near to the ice shelf terminus that led to the calving of a large tabular iceberg in February 2020 (Fig. 1 a2) - part of a series of calving events regarded to have had significant consequences for the dynamics of Pine Island Glacier (Joughin et al., 2021). Finally, there are a large number of fractures on the southern shear margin of Pine Island Ice Shelf (Fig. 1 a3).


We create the strain-rate contribution to $\xi$ using the same velocity data that we use in our misfit function. To estimate the derivatives $\partial_i u_j$, we differentiated the velocity components using a method described in Chartrand (2017), using Tikhonov regularisation to promote smoothness (regularisation parameters were chosen with some trial-and-error, where preference was given to solutions in which regions of high shear varied smoothly over lengthscales comparable to the widths of visible

shear margins). Aligning the $x$-coordinate with local flow direction, we define regions of *high shear* to be those in which $|\dot{\varepsilon}_{xy}| > 0.1 \text{ a}^{-1}$. This threshold is a bit discretionary, though it corresponds to stresses within the range $90 - 320 \text{ kPa}$ of tensile strength suggested in Vaughan (1993) for a wide range of englacial temperatures. Then $\xi_{\text{shear}} = \max\{0, 1 - 10|\dot{\varepsilon}_{xy}|\}$ (Fig. 1 c) and $\xi = \min\{\xi_{\text{frac}}, \xi_{\text{shear}}\}$. This data picks out the shear margins of the glacier, as well as the velocity discontinuity associated with the rift close to the ice shelf calving front.


In the case of the snapshot inverse problem, the assumption we wish to encode in our prior for $\phi$ is that $\phi \to 1 + \epsilon$ as $\xi \to 1$, where $\epsilon \sim \mathcal{N}(0, \gamma^2)$ and $\gamma$ is a small number. Such a prior over the $\phi$ field can be written:

$$p_\Phi(\phi) \propto \exp(-\frac{1}{2\gamma^2} \int_\Omega (1 - \phi)^2 \xi \, d\Omega). \tag{6}$$

Assuming the distribution of measurement errors is isotropic, with covariance $\sigma^2 \mathcal{I}$, this translates to a regularisation term:

$$\alpha_\phi = \frac{\sigma^2}{\gamma^2}, \quad \mathcal{J}_\phi(\phi) = \int_\Omega (1 - \phi)^2 \xi \, d\Omega. \tag{7}$$

A greater exposition of this link between priors and regularisation parameters is given in appendix A.

We solve the inverse problem for the regularisation term shown in (7), as well as the heuristic regularisation (5) and no regularisation.




The initial guess for the control fields can have a large influence on the optimisation problem. For the $\phi$ field, we use an initial guess of 1 everywhere. For $C$, we take the view that the initial guess should be the field required to reproduce the observations on grounded ice as closely as possible with a uniform $\phi = 1$. Hence, before carrying out the full optimisation including both control fields, we solve an inverse problem for $C$ with fixed $\phi = 1$, matching speeds only on grounded ice and use this as the initial guess. This has the effect of considerably reducing the deviation of $\phi$ from 1 in the solution. This has the added bonus of allowing us to search independently for the regularisation parameters $\alpha_C$ and $\alpha_\phi$.

## 2.2 Fracture data assimilation through time

As previously noted, the field $\phi$ contains contributions from sources that cannot easily be distinguished by the spatial scales on which they vary. However, it seems likely that the contribution to ice softness due to fracturing varies on a shorter *temporal* scale than any other contribution. Hence, while attributing ice softness to the presence of fractures requires a large number of assumptions, we can reasonably attribute changes in ice softness required by the model to fit observations over monthly-annual timescales to the fracturing or 'healing' of ice.

Given a series of timesteps with times $\{t_i | i = 1, ..., n\}$, separated by $\Delta t$ (e.g. one month), we solve the following inverse problem for the control parameters $(C_i, \phi_i)$ at each timestep:

$$(C_i, \phi_i) = \underset{C_i, \phi_i}{\mathrm{argmin}}\{\mathcal{J}_m(u_i, u_{o_i}) + \alpha_C \mathcal{J}_C(C_i) + \alpha_\phi \mathcal{J}_\phi(\phi_i) + \frac{\alpha_t}{\Delta t}\mathcal{J}_t(\phi_i, \phi_{i-1})\}, \tag{8}$$

where we have introduced the regularisation through time $\mathcal{J}_t(\phi_i, \phi_{i-1})$ relating the softness in the current timestep to that of the previous timestep. Though not particularly sophisticated, a method such as described by Eq. (8) is immediately amenable to the introduction of fracture data in the form of $\mathcal{J}_t$. Previous studies (Hogg et al., 2017; Selley et al., 2021) have used such a method with $\mathcal{J}_t = \int_\Omega |\phi_i - \phi_{i-1}|^2 d\Omega$ and we modify this only slightly here. We propose the regularisation function:

$$\mathcal{J}_t = \int\limits_\Omega (1 - |f_i - f_{i-1}|) \times |\phi_i - \phi_{i-1}|^2 d\Omega \tag{9}$$

where $f_i$ is the map showing the locations of fractures over the domain at time $t_i$. Hence, changes to the softness field are preferred in regions in which the fracture pattern has changed, with a strength that depends on the length of the timestep and the regularisation parameter $\alpha_t$. For these problems, we also set $\alpha_\phi = 0$.

We carry out such a procedure on Pine Island Glacier with 5 years of speed and fracture observations from December 2016 to December 2021, and timesteps of one month. This captures three calving events and the major disintegration of the southern shear margin of the ice shelf, and that of the calving front of Piglet Glacier (Joughin et al., 2021; Surawy-Stepney et al.,





2023b). For each month, we use the mean speeds measured over that month as our observed speeds, and median fracture map composites.

## 3   Results

### 3.1   Snapshot inverse problems

We begin with the results of fracture data assimilation applied to a snaphot inverse problem on Pine Island Ice Shelf. We
consider how using the data-informed regularisation alters the problem compared to a case of no regularisation, and the heuristic regularisation of eq. (5). We refer to optimisations in which $\phi$ is unregularised as 'case 1', those in which we apply heuristic Tikhonov regularisation as 'case 2' and those in which we apply the data-informed regularisation given by eq. (7) as 'case 3'. We look at the misfits, the output control fields and changes to the problem conditioning.

#### 3.1.1   Softness fields

The $\phi$ fields in each case are substantively different on Pine Island Glacier for this set of geometry and speed data (Fig. 2). This is true for both the grounded and floating ice. Firstly, in both cases 1 and 2 there are large deviations of $\phi$ from 1 far upstream of the grounding line including substantial softening in the shear margins of even slow-flowing ice streams (Fig. 2 a, b). This is completely absent in the solution to case 3 (Fig. 2 c). Given the lower misfits in these regions (Fig. 2 e, f) compared to case 3 (Fig. 2 g), it appears that the model finds it difficult to compensate for the velocity gradients at the margins of the tributary
ice streams by enhancing gradients in $C$ where it is encouraged not to alter $\phi$. In the large fractured region upstream of the grounding line (Fig. 1 a, b), the solution for case 3 shows higher amplitude deviations of $\phi$ from 1 than in cases 1 and 2.

The differences in $\phi$ between the different forms of regularisation are just as pronounced on the floating ice shelf. In cases 1 and 2, softnesses on the ice shelf are smooth and spread to large distances either side of the shear margins. In contrast, in
the solution to case 3, softness is concentrated in the shear margin with larger amplitude deviations of $\phi$ from 1 confined to a smaller area. A portion of the solution degeneracy for $\phi$ on Pine Island Glacier occurs because the central shelf moves almost entirely by pure advection. In the absence of any significant strain rates, most solutions for $\phi$ in this region fit the data equally well. The inclusion of an explicit prior appears to help with this by encouraging stiff ice on the central shelf.

The rift that propagated across the ice shelf at the time the speed data was collected caused a discontinuity in the data. The feature is much more clearly resolved in the solution to case 3 than case 2, and even case 1. Hence, it appears difficult for the model to assign low values of $\phi$ to a region very local to the rift unless encouraged to do so. This is perhaps due to the distributed influence of the ice at the terminus on the dynamics of the ice shelf as a whole (Joughin et al., 2021; Bevan et al., 2023). On the floating ice, the misfit for case 3 is considerably better than case 2 (Fig. 2 e-f).



**Figure 2.** Solutions to the inverse problem with three methods of regularisation. a-c) Stiffness fields for the unregularised, heuristically regularised and data-informed inverse problems respectively. d-f) Misfits for the unregularised, heuristically regularised and data-informed inverse problems respectively. Background images are the MODIS Mosaic of Antarctica (Haran et al., 2021), and grounding lines (shown in black) are according to Rignot et al. (2016).

### 3.1.2 The effect on problem conditioning

A well conditioned problem damps the contribution of oscillatory, high frequency components of the input data, such as uncorrelated noise in the measured speed, while an ill-conditioned problem is highly sensitive to it. Bringing prior information into the inverse problem has the potential to change the conditioning by enhancing gradients in previously flat regions of the cost landscape. In order to test this change in conditioning, we investigated the impact of perturbations in the input velocity data on





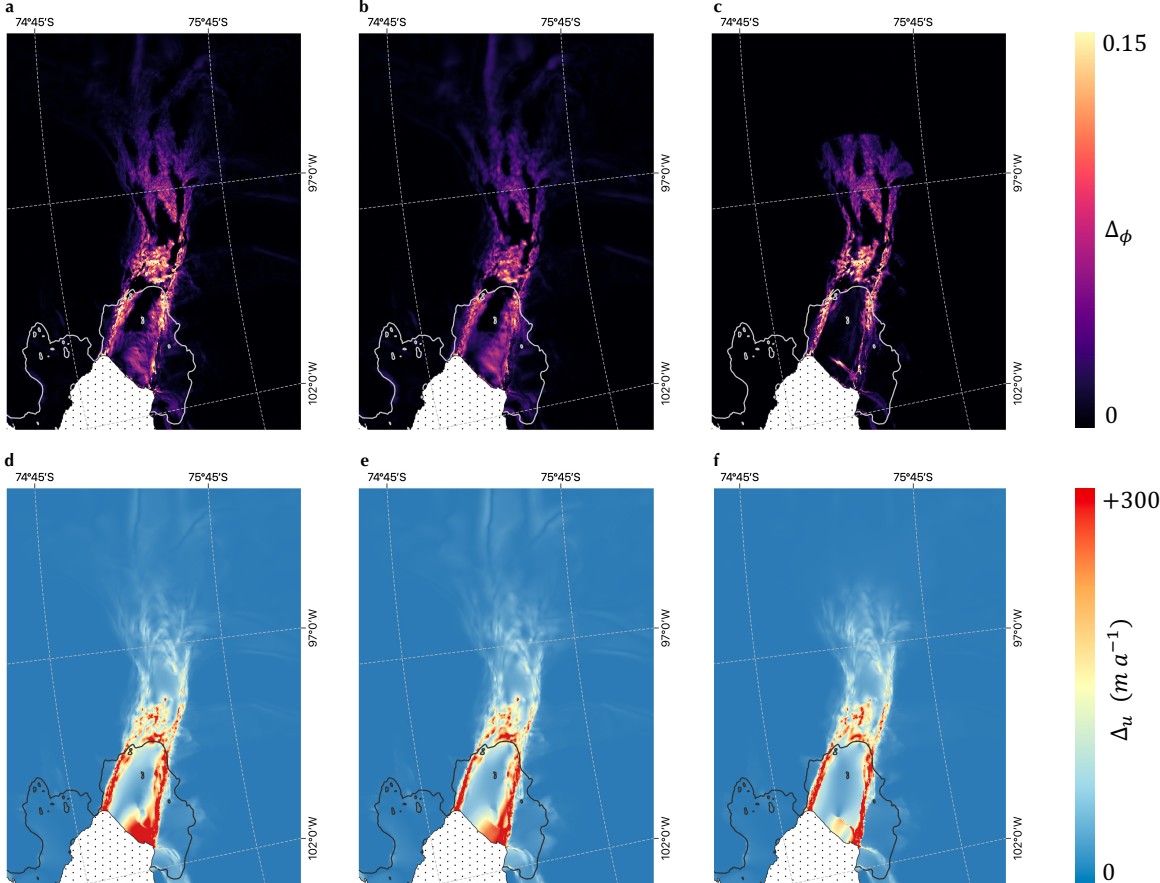

**Figure 3.** Variation in the solutions for the three methods of regularisation. a-c) Standard deviation in the softness fields between 10 optimisations with Gaussian noise added to the speed data for the unregularised, heuristically regularised and data-informed inverse problems respectively. d-f) Associated standard deviations in the modelled speed for the unregularised, heuristically regularised and data-informed inverse problems respectively.

the spread of resulting $\phi$ and $u$ fields.

We performed 10 inverse problems with the addition of uncorrelated Gaussian noise to the input data for the case of data-informed regularisation, heuristic regularisation and no regularisation. Noise was added with a mean of zero and standard deviation of 10% of the local speed. In each case, we measured the cell-wise standard deviation over the 10 $\phi$ and $u$ output

fields (Fig. 3).

Unsurprisingly, the regularised problems show a smaller spread in the solutions for the control fields - suggesting improved conditioning (Fig. 3 a-c). The spread of solutions for $\phi$ is confined in the case of the data-informed regularisation to the regions of very low $\xi$, while in those regions, the standard deviations are of similar magnitude to the unregularised case. This is expected



because in essence, the data-informed regularisation separates regions in which high-amplitude deviations of $\phi$ from from 1
are penalised (where $\xi \to 1$) from regions that are entirely unregularised. The heuristic regularisation, case 2, that is explicitly
devised to improve the problem conditioning indeed looks to result in the most well-conditioned problem on grounded ice.
However, this is not the case on the central ice shelf, where the degeneracy described above leads to a larger solution variance
than in the data-informed case. The spreads of speed (Fig. 3 d-f) reflect the spreads of the control fields.

**3.2 Inverse problems through time**

We consider two instances of temporal regularisation of the type described in eq. (8): the 'data-informed' case:

$$\mathcal{J}_t = \int_\Omega (1 - |f_i - f_{i-1}|) \times |\phi_i - \phi_{i-1}|^2 d\Omega \quad \text{and } \alpha_\phi = 0,\ \alpha_t = 5 \times 10^6, \tag{10}$$

and the 'heuristic' case:

$$\mathcal{J}_t = \int_\Omega |\phi_i - \phi_{i-1}|^2 d\Omega \quad \text{and } \alpha_\phi = 1.5 \times 10^9,\ \alpha_t = 10^4, \tag{11}$$

equivalent to that used in Selley et al. (2021).

Using fracture data in successive timesteps to weight the temporal regularisation has a significant effect on the softness
fields over the five years of observations compared with the simpler approach (Fig. 4 a, b). The data-informed case leads to
features of low $\phi$ which resemble crevasses starting to appear in the southern shear margins after $\sim 18$ months (black dotted
arrow Fig. 4 b). Rifts that led to the calving of large icebergs in October 2018 and February 2020 are visible as highly linear
features of soft ice in the solutions to the data-informed problem (black dashed arrows Fig. 4 b). These features are visible in
Fig. 4 a, though are less easily discernible as rifts. The softness fields in the two cases appear similar by May 2021, with that
of the heuristic regularisation looking essentially like a blurred out version of the data-informed case. Both show the southerly
migration of the seaward end of the southern shear margin through the time period, and, by 2021, a stripe of soft ice that
connects the shear margins of Pine Island and Piglet Ice Shelves. It is only clear in Fig. 4 b (black solid arrow) that this stripe
of soft ice corresponds to a number of long, parallel rifts. Diffuse blobs of softness can be seen on the central ice shelf in Fig. 4
a (May 2021, grey arrow) which are not present in the data-informed case. As the simulation contains no thickness advection
and no accumulation rate is specified, it is possible that these, which are not present in the data-informed case, could be the
result of localised thinning. Otherwise they could once more be the result of under-determinedness. This latter possibility is
perhaps more likely given how agnostic the model is to the values of $\phi$ in the central trunk and that the gravitational forcing is
not modified by a change in stiffness.

Throughout the simulation period, the misfits associated with each case are very similar, with generally slightly larger mean
misfits over the region in the data-informed case (Fig. 4 c, d). The exceptions to this are in the months in which calving





**Figure 4.** The evolution of the stiffness on Pine Island Ice Shelf between June 2018 and May 2021 for heuristic (a) and data-informed (b) regularisation. c) Mean misfit over the ice shelf for the two cases through time. d) Mean misfit over the ice shelf for the heuristically-regularised problem. e) Timeseries of mean misfit over the ice shelf for the data-informed and heuristically-regularised problems. Background images in a and b are the MODIS Mosaic of Antarctica (Haran et al., 2021), and grounding lines (shown in black) are according to Rignot et al. (2016).

events occur - where the misfit is generally elevated as the model struggles to deal with the sudden appearance of large velocity gradients near the glacier terminus. At these times, the data-informed case does slightly better as the observations of rift growth nudge the model towards the right pattern of softening near the terminus.





## 4 Discussion

The problem of accurately estimating ice softness and basal slip fields from observations of ice speed is dogged by the spector
of ill-posedness. In an effort to improve this, we have presented two simple ways of assimilating fracture data into the inverse
problem for a marine-terminating ice stream, as a way of providing the problem with genuine prior information. In a number
of ways, the effect of these methods, their success and what we learn from the experiments carried out in this study differ for
grounded and floating ice, so we first review these separately.

### 4.1 Grounded ice

As discussed above, the presence and evolution of fractures is only a contributing factor in determining $\phi$, and the efficacy of
the methods presented here depend on the extent to which we apportion softness to fracturing. We have seen in our example
of snapshot problems over Pine Island that softness fields on grounded ice found using the data-informed regularisation vary
considerably within contiguous areas of observed fracture (Fig. 2 c). If fracturing in these regions were truly the main contrib-
utor to ice softness, one would expect $\phi$ to be uniformly less than 1 this region - visually mimicking the uniform coverage of
the region by surface fractures (Fig. 1 a1). This suggests that here at least, the dominant contribution to our uncertainty in the
material properties of the ice softness is not the unaccounted for presence of fractures, but some combination of other factors.
This is consistent with the fact that prescribing the data-informed regularisation on the grounded ice dampens the softness away
from these regions of fracture but does not change the shape of the solution greatly within them. This suggests that observations
of surface fracture on grounded ice have limited use in reducing the degeneracy between solutions caused by overlapping $C$
and $\phi$ fields, directly answering in the negative a suggestion made in Surawy-Stepney et al. (2023b).

In addition, this constitutes a certain amount of evidence that this kind of grounded surface crevasse has a limited impact
on ice dynamics, despite the very low basal frictions we find in this part of Pine Island Glacier (Joughin et al., 2009) and the
enhanced membrane stresses required to compensate for this. This is consistent with previous assumptions that the depths of
these crevasses is only a small fraction of the ice thickness (Benn and Evans, 2014).

Finally, it is worth noting that the softness fields on grounded ice (and also substantially on floating ice) found using heuristic
regularisation (Fig. 2 b) mimic many of the features of the strain rate map in Fig. 1 c. This suggests greater potential for this
data to be used to constrain the softness and that the prior we are currently using doesn't fully capture our assumption that
softness should be related to shear (as that of Ranganathan et al. (2021) might, for example). A better prior might, for example,
be to assume softness is linear in princpal strain rate. Future work should look to investigate different priors that better utilise
the strain rate data at our disposal.





## 4.2 Floating ice

We have shown in both snapshot inverse problems and time-dependent inverse problems that the softness fields over floating
ice, resulting from use of our proposed regularisation methods, appear more like what we would expect if the softening were
due to fracturing/shearing compared to more heuristic regularisation methods. When encouraged to do so, the model is happy
to concentrate softness in regions of observed fracture or high shear without suffering a worse misfit with the prescribed speed
data. It is tempting to think that this results in softness fields that appear more likely to accurately represent the material
properties of the ice shelf at the time the ice speed data was collected. Unfortunately, the ill-posedness of the problem means
that methods of evaluating whether this is true do not extend far beyond a visual assessment of whether the solutions 'look
right' in the context of our priors, however this is a technique that should not be ignored! Though the correlation between
rheological parameters, inferred in a manner similar to that described in the heuristic regularisation case here, and crevasse
data has previously been shown to be limited (Gerli et al., 2024), we have shown in both the snapshot and time-dependent
cases that there are solutions to the inverse problem with at least equally good misfit in which this correlation is undoubtedly
strong.

### 4.2.1 When would we use these methods?

The example we have chosen for the snapshot inverse problem, where a large rift can be seen on the central trunk of Pine Island
Ice Shelf along with an associated discontinuity in $u_o$, is somewhat contrived to show the differences between the regularisation
methods discussed. It is unlikely that a model-user looking to initialise a century-long simulation would choose such data, and
would do better to choose data from a time more representative of a typical state of the glacier. Even if a typical state does
include fractures and speed discontinuities, without a method of sensibly evolving the softness field through time, it would be
reasonable to initialise a model with a smoother solution for $(C, \phi)$ that might be less representative of the true initial state, but
is also less *specific* to it. Hence, softness fields found with the use of fracture data and regularisation procedures we propose
here are more likely to be useful in diagnostic simulations, or transient simulations with timescales on the order of years.


A major motivation for investigating these methods of constraining the inverse problem is that the time-varying solutions
have potential use in evaluating models that take a continuum damage mechanics approach to parameterising the effect of
fractures on large-scale ice rheology (e.g. Sun et al. (2017)). In particular, the softness fields shown in Fig. 4 b could be used to
constrain the way in which a scalar damage field, that acts isotropically on the rheology, is evolved by such a model (Borstad
et al., 2016).

## 4.3 A note on L-curves

Fig. 5 shows, on a logarithmic scale, solution and misfit norms at convergence for a number of possible regularisation pa-
rameters $\alpha_\phi$ for eq. (7), known as an L-curve (Hansen and O'Leary, 1993). Intuition suggests that one should choose the




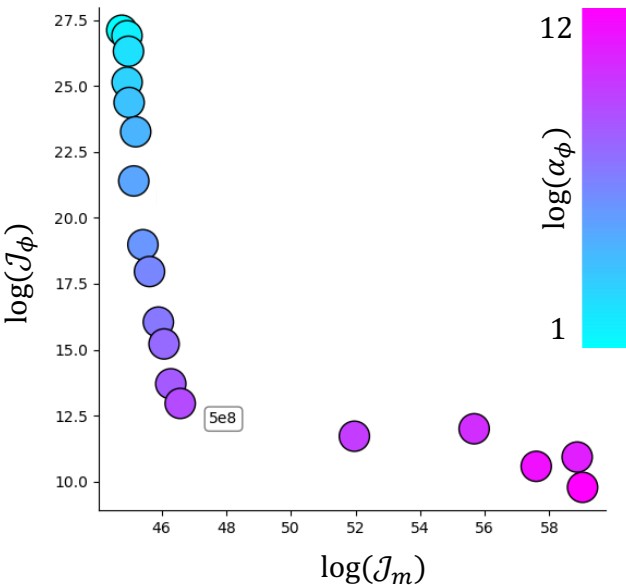

**Figure 5.** L-curve for the data-informed regularisation. Solution norm (y) and misfit (x) are plotted on a logarithmic scale for different choices of the regularisation parameter $\alpha_\phi$.

regularisation parameter at the corner of the L-curve, which balances the regularisation and misfit components of the cost function. This can be shown in some circumstances to be the point at which contributions to the solution are balanced between errors in the data and errors in the regularisation (Hansen, 2000). In our case this is $\alpha_\phi \approx 5 \times 10^8$. However, this choice of parameter results in solutions with fewer crevasse features than we expect to see - such as the rift near the ice shelf terminus (Fig. 4 b). Hence, in practise, we choose a parameter an order of magnitude smaller, where we are satisfied with the misfit (staying on the 'vertical branch' of the L-curve) but can see some of the detail we believe should be present in the softness field. Though very useful, L-curve analysis can be a blunt instrument and should always be used alongside other heuristics such as visual assessment of the control fields in deciding the regularisation parameter. Its use is based on the assertion that the preferred solution to an inverse problem is one that contains the least extraneous structure (Wolovick et al., 2023). However, for structure to be deemed 'extraneous', a cost function that encodes a good deal of your prior knowledge is required, which is not often available. This tendency for L-curve analysis to produce over-regularised solutions has been noted previously (e.g. Chamorro-Servent et al. (2019); Milovic et al. (2021), and notably in Recinos et al. (2023)).

## 5 Conclusions

We have introduced two ways in which fracture location data, and in one case strain rate data, can be used as prior information to inform the estimation of basal slip and ice softness fields from observations of ice speed. Applications of these methods to snapshot and time-dependent inverse problems over Pine Island Glacier show that little is gained in their use compared to the


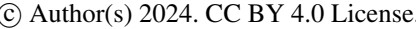

use of popular heuristic regularisation methods when considering the solutions on grounded ice. This suggests that a failure to account for the presence fracturing does not dominate our uncertainties in the material properties of grounded ice. This is not true, however, on floating ice, where we see the resolution of fracture features in the static and time-varying softness fields without impacting the misfit, and a reduction in solution degeneracy in regions of low strain rates. This suggests that such

methods can be used to provide us with softness fields that better represent the true material properties of the ice shelf at the time of the acquisition of the ice speed data. Such softness fields have potential use in diagnostic modelling, and in constraining models seeking to evolve softness fields in time.

*Acknowledgements.*  The authors gratefully acknowledge the European Space Agency (ESA) and the European Commission for the acquisition and availability of Copernicus Sentinel-1 data. Funding is provided by ESA via the ESA Polar+ Ice Shelves project (ESA-IPL-POE-

EF-cb-LE-2019-834) and the SO-ICE project (ESA AO/1-10461/20/I-NB) to AEH which both are part of the ESA Polar Science Cluster. Funding is provided from NERC via the DeCAdeS project (NE/T012757/1) and the UK EO Climate Information Service (NE/X019071/1) to AEH

*Code availability.*  The BISICLES Ice Sheet Model is open source and the code is available at: https://commons.lbl.gov/display/bisicles/BISICLES. Additional code required to run the simulations in this study can be found at https://zenodo.org/doi/10.5281/zenodo.13694744.

*Data availability.*  The authors have made available, for the purposes of review, data used in the modelling presented in this article at https://zenodo.org/doi/10.5281/zenodo.13694744. This includes geometry and speed data, as well as the priors, based on fracture and strain rate maps, used for the snapshot and 'time-dependent' inverse problems.

*Author contributions.*  TSS and SLC designed the work. TSS carried out the model development and simulations and wrote the manuscript. All authors contributed to discussion.

*Competing interests.*  The authors declare that they have no conflict of interest.





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





## Appendix A: Deriving a regularisation term from a prior distribution

470 Consider a version of the inverse problem in which the $C$ field is known. Then the forward model solves $u(x) = f(\phi(x))$. We can replace this with the stochastic model:

$$U = f(\Phi) + E \tag{A1}$$

where $U$, $\Phi$ and $E$ are random variables representing modelled speed, the stiffness $\phi$ and an additive error respectively (Calvetti and Somersalo, 2018). Let the error be governed by the probability distribution $p_E$. Eq. (A1) then implies:

475 $$p_{U|\Phi}(u|\phi) = p_E(u - f(\phi)) \tag{A2}$$

and Bayes' rule gives:

$$p_{\Phi|U}(\phi|u) \propto p_E(u - f(\phi))p_\Phi(\phi), \tag{A3}$$

where $p_\Phi(\phi)$ constitutes our prior for the distribution of $\Phi$.

480  We assume that as $\xi \to 1$, $\phi \to 1 + \epsilon$, where $\epsilon \sim \mathcal{N}(0, \gamma^2)$. We can encode this as the following relation for $\Phi$:

$$\xi(1 - \Phi) = \gamma W \tag{A4}$$

where $\gamma$ controls how much we allow $\Phi$ to vary, and $W$ is a Gaussian random field with zero mean and identity covariance. Then

$$p_\Phi(\phi) \propto \exp\left(-\frac{1}{2\gamma^2} \int_\Omega (1 - \phi)^2 \xi^2 \, d\Omega\right). \tag{A5}$$

  Assuming Gaussian error distribution with zero mean and isotropic covariance $\sigma^2 \mathcal{I}$, gives

485 $$p_E(u - f(\phi)) \propto \exp\left(-\frac{1}{2\sigma^2} \int_\Omega (u - f(\phi))^2 \, d\Omega\right) \tag{A6}$$

  Hence, from eq. (A3):

$$p_{\Phi|U}(\phi|u) \propto \exp\left\{-\frac{1}{2\sigma^2}\left(\int_\Omega (u - f(\phi))^2 \, d\Omega + \frac{\sigma^2}{\gamma^2} \int_\Omega (1 - \phi)^2 \xi \, d\Omega\right)\right\}, \tag{A7}$$

making the assumption that $\xi^2 \approx \xi$.

  A maximum a posteriori estimate for $\phi(x)$ given $u(x)$ is, therefore, the solution to:

490 $$\phi_{MAP} = \arg\min\left\{\int_\Omega (u - f(\phi))^2 \, d\Omega + \frac{\sigma^2}{\gamma^2} \int_\Omega (1 - \phi)^2 \xi \, d\Omega\right\}, \tag{A8}$$



i.e. a minimisation over our original cost function with:

$$\alpha_\phi = \frac{\sigma^2}{\gamma^2}, \quad \mathcal{J}_\phi(\phi) = \int_\Omega (1-\phi)^2 \xi \, d\Omega. \tag{A9}$$

A reasonable prior might be to allow $\phi$ to vary from 1 away from fractured areas ($\xi \to 1$) with a standard deviation of 0.1, corresponding to $\gamma^2 = 0.01$. Taking $\sigma$ to be of order 100 m/y, this gives us a value of $\alpha_\phi \sim 10^6$ for the coefficient of the $\phi$ regularisation term in our initial
cost function. Note, we have assumed in this analysis a spatially uniform estimate of uncertainty in our velocity observations. If a more reliable estimate of this uncertainty existed, it could be included as a modification to $\xi$.

## Appendix B:  Derivatives of $\mathcal{J}_\phi$

The inverse problem is solved using a nonlinear conjugate gradient method. This requires the projection of the Jacobian $\nabla \mathcal{J}(\phi)$ along the
direction of the residual $u - u_o$.

Let $\phi = \phi_0 e^q$ so that $\phi > 0$.

Define the stiffness part of the cost function as:

$$\mathcal{J}_\phi = \int_\Omega (1-\phi)^2 \xi d\Omega \tag{B1}$$

The Gâteaux differential is defined by the projection of the functional gradient onto the direction defined by a perturbation $\delta q$:

$$\langle \delta \mathcal{J}_\phi, \delta q \rangle = \lim_{\epsilon \to 0^+} \frac{\mathcal{J}_\phi(q + \epsilon \delta q) - \mathcal{J}_\phi(q)}{\epsilon}, \tag{B2}$$

where the binary operator $\langle \cdot, \cdot \rangle$ is the inner product over the space of functions. In our case:

$$\langle \delta \mathcal{J}_\phi, \delta q \rangle = -\int_\Omega \delta q \phi (1-\phi) \xi d\Omega \tag{B3}$$

and we interpret $\phi(1-\phi)\xi$ as the functional gradient. This is calculated in each iteration of the non-linear conjugate gradient method, and is used to update the value of $q$.