# Peer review of "Using observations of surface fracture to address ill-posed ice softness estimation over Pine Island Glacier"

_EGUsphere, 2024_

## Referee Comment (RC1)

Review of "Using observations of surface fracture to address ill-posed ice softness estimation over Pine Island Glacier" by Surawy-Stepney et al.

In this manuscript, the authors investigate the effect of assimilating more prior information into inversions for ice stiffness. The data informing the priors are strain rates, and locations of fracture derived from satellite imagery. Pine Island is chosen as a study area due to the large amount of fracturing observed there. Experiments are carried out using both snapshot and time-dependent inversion processes, using different regularisation. The results show that the use of this data in priors results in stiffness fields which better visually represent observed fracture patterns, without affecting the velocity misfit. The use of methods informed by fracture data could be important for improving inversions of floating ice, but is likely not have much impact on grounded areas. It is suggested that these methods would be best suited to diagnostic modelling and attempts to evolve stiffness fields through time.

This study will be valuable to a particular niche of ice flow modellers, and is certainly within the scope of The Cryosphere. I personally found it to be interesting, although I think wider interest will be limited as the focus is only on the inversion process and, by the authors' own admission, unlikely to be of much help to long-term predictive simulations.

My main issue with this manuscript is that it can be quite difficult to read, and is unclear at times. The introduction seems a little muddled, with some parts referencing specifics of this study among a more general review of the relevant issues. I would recommend moving anything specific to this study (sliding law, value of $n$ in flow law etc.), and the more detailed discussion of reasoning behind the methods used found in the last paragraphs, into the methods section, so that it can all easily be found and doesn't over-complicate the introduction.

I also found the methods section difficult to follow in places. Section 2.1 would in my opinion benefit from being restructured. I also think the methods section should contain a clear summary of all the experiments which were carried out, as these are not all introduced until during the results section.

The scientific content of this manuscript is good, and worthy of publication, but I think work needs to be done to improve the clarity of its presentation. For this reason, I recommend publication after revisions.

Specific Comments

Line 27: Is this a typo, or is the approximately equal sign there for a reason? If not a typo, please explain what is meant and be clear what value for n is being used in your work.

Line 30: It may be helpful to write Eq.1 in a form which includes φ(x) for clarity.

Line 43: I think the sliding law used in this study should be stated in the methods section rather than the introduction.

Lines 48-51: It's a little unclear at points in this introduction whether you are talking about the specific process(es) used in your study, or more generally. As a more general point, some inversion

processes use $u$ and $v$ velocity components as two separate observed fields, and some can also make use of thickness changes ($dh/dt$). This doesn't mean the problem is ever not ill-posed, but there is a greater variety in approaches that just using a single $u$ field. If this statement is referring to the specific process used in this study, please make this clear.

Lines 92-5: This detail probably belongs in the methods section.

Lines 97-102: As above, better to put the detail in methods.

Lines 113-4: Could this point about the link between dynamics and fracturing over the rest of Antarctica be expanded on in the discussion?

Lines 121-2: You refer to this past paper a few times without detail. As it relates to an important source of data in this study, a brief description of the method would be useful in this section, or at least mentioning that it uses a machine learning technique to identify crevasses.

Lines 128-49: In my opinion, these paragraphs would benefit from a little restructuring. I think the definition of $\xi = \min\{\xi_{frac}, \xi_{shear}\}$ should be introduced first, defining what the components are, before then presenting the details of how the components are calculated. This would have made it easier for me to follow, although that may be a personal preference.

Lines 161-3: Could you give a reason for the choice of initial guesses? After stating that this can have a large influence on the optimisation, I feel a justification of the choice is required. Why not, for example, use a uniform guess for C or a value of 0.5 for $\phi$?

Lines 194-7 (also Lines 226-30, 241-245): I think a summary of all experiments should be included at the end of the methods section, before going into the results. This will help to show readers exactly what you're doing in the context of methodology you've described. Introducing the exact cases during the results section seems a bit late.

Lines 203-4: The subpanel letters do not match the figure. These should be d,e,f not e,f,g.

Lines 283-5: This is worded quite vaguely. If a reference to the previous paper is required (I would argue it is not here), be clear about what suggestion is being referred to.

Lines 334-6: The chosen value should also be labelled on Fig.5. In fact, it would be good to have the values labelled for each circle on the figure.

---

## Author Comment (AC1)

**Responses to reviewer comments for the article "Using observations of surface fracture to address ill-posed ice softness estimation over Pine Island Glacier"**

We would like to thank the editor and reviewers very much for the taking the time to read the article and for providing us with valuable and insightful feedback. All reviewer comments and responses are collated in this document. Each review is reproduced here in full. Responses to any general comments of the reviewers are coloured in teal, while responses to specific comments are tabulated afterwards.

A central theme of both reviews is that parts of the article should be restructured to make it easier to follow. To this end, I have made various structural changes, informed by specific comments made by the reviewers, for example, moving text between the introduction and methods sections. We hope the reviewers find that these changes have improved the flow and clarity of the article.

**Responses to comments from Reviewer #1**

**Reviewer 1:** In this manuscript, the authors investigate the effect of assimilating more prior information into inversions for ice stiffness. The data informing the priors are strain rates, and locations of fracture derived from satellite imagery. Pine Island is chosen as a study area due to the large amount of fracturing observed there. Experiments are carried out using both snapshot and time-dependent inversion processes, using different regularisation. The results show that the use of this data in priors results in stiffness fields which better visually represent observed fracture patterns, without affecting the velocity misfit. The use of methods informed by fracture data could be important for improving inversions of floating ice, but is likely not have much impact on grounded areas. It is suggested that these methods would be best suited to diagnostic modelling and attempts to evolve stiffness fields through time.

This study will be valuable to a particular niche of ice flow modellers, and is certainly within the scope of The Cryosphere. I personally found it to be interesting, although I think wider interest will be limited as the focus is only on the inversion process and, by the authors' own admission, unlikely to be of much help to long-term predictive simulations.

My main issue with this manuscript is that it can be quite difficult to read, and is unclear at times. The introduction seems a little muddled, with some parts referencing specifics of this study among a more general review of the relevant issues. I would recommend moving anything specific to this study (sliding law, value of n in flow law etc.), and the more detailed discussion of reasoning behind the methods used found in the last paragraphs, into the methods section, so that it can all easily be found and doesn't over-complicate the introduction.

I also found the methods section difficult to follow in places. Section 2.1 would in my opinion benefit from being restructured. I also think the methods section should contain a clear summary of all the experiments which were carried out, as these are not all introduced until during the results section.

The scientific content of this manuscript is good, and worthy of publication, but I think work needs to be done to improve the clarity of its presentation. For this reason, I recommend publication after revisions.

**Response:** We would like to thank the reviewer for their positive comments about the article and their thorough and thoughtful review. I have restructured the introduction and methods sections (and the results to some degree) to make things clearer. Many of the changes made are in response to specific comments laid out in the review below.

**Responses to specific comments from Reviewer #1**

| | Reviewer 1 | |
|---|---|---|
| ID | Reviewer Comment | Response |
| 1 | Line 27: Is this a typo, or is the approximately equal sign there for a reason? If not a typo, please explain what is meant and be clear what value for n is being used in your work. | This is not a typo, though I have made this less ambiguous by adding the sentence: "The value of the exponent $n$ is dependent on the particular mechanisms by which creep occurs within the ice and various properties of the crystal grains (e.g. Haefeli (1961)), and takes a value between 1 and 4 in most cases. Here, we take the common reference value of $n = 3$". |
| 2 | Line 30: It may be helpful to write Eq.1 in a form which includes $\phi(x)$ for clarity. | I have changed this sentence to be: Here, we consider the approach in which these are prescribed *a priori* and a 'stiffness' field $\phi(x)$ is defined over the domain to account for unknown deviations in the expected ice rheology, such that eq. 1 becomes $\tau_{ij} = 2\phi\eta\dot{\varepsilon}_{ij}$ |
| 3 | Line 43: I think the sliding law used in this study should be stated in the methods section rather than the introduction | Thank you, I have moved this line into the methods. |
| 4 | Lines 48-51: It's a little unclear at points in this introduction whether you are talking about the specific process(es) used in your study, or more generally. As a more general point, some inversion processes use $u$ and $v$ velocity components as two separate observed fields, and some can also make use of thickness changes $dh/dt$. This doesn't mean the problem is ever not ill-posed, but there is a greater variety in approaches that just using a single $u$ field. If this statement is referring to the specific process used in this study, please make this clear. | I hope that the changes made to the introduction and methods have addressed the issue of clarity and distinguished statements that relate to methods in general and those we use ourselves in the article. I have also added a couple of sentences that make it clear that other data can be included in the inverse problem as suggested. |
| 5 | Lines 92-5: This detail probably belongs in the methods section | This has been moved into the methods. |
| 6 | Lines 97-102: As above, better to put the detail in methods. | This has also been moved into the methods |

| 7 | Lines 113-4: Could this point about the link between dynamics and fracturing over the rest of Antarctica be expanded on in the discussion? | I appreciate the desire to expand on this, but I think it might be difficult to do so without moving into speculation. PIG has shown a strong connection between fracturing and broader dynamics over the last decade, e.g. the cited studies showing links between calving, shear margin degradation and changes in ice speed; also the very low basal stresses found far inland of the grounding line. There is a feeling that this is not replicated in other places round Antarctica, though there is actually little concrete evidence of that. For example, when carrying out inverse problems, I have not seen very low basal stresses on grounded ice in many other places, but I haven't actually done or seen proper analysis on it. I think it might be better to keep this to a short statement reflecting a generally held belief than include too much pontification. I could be persuaded otherwise though. |
|---|---|---|
| 8 | Lines 121-2: You refer to this past paper a few times without detail. As it relates to an important source of data in this study, a brief description of the method would be useful in this section, or at least mentioning that it uses a machine learning technique to identify crevasses. | I have included a short paragraph with a little more detail about this dataset. |
| 9 | Lines 128-49: In my opinion, these paragraphs would benefit from a little restructuring. I think the definition of $\xi = \min\{\xi_{\text{frac}}, \xi_{\text{shear}}\}$ should be introduced first, defining what the components are, before then presenting the details of how the components are calculated. This would have made it easier for me to follow, although that may be a personal preference. | Thank you for the good suggestion, I have changed the structure as suggested. |

| 10 | Lines 161-3: Could you give a reason for the choice of initial guesses? After stating that this can have a large influence on the optimisation, I feel a justification of the choice is required. Why not, for example, use a uniform guess for $C$ or a value of 0.5 for $\phi$? | This is a very good point! The optimisation problem will be more easily solved if the initial guess is close to the optimal solution. If we think Glen's flow law with a choice of $n = 3$ is correct, the ice is unbroken, and the temperature field we get from the thermomechanical spin-up described in the text is a good approximation, then taking $\phi = 1$ is the right choice. Even if those assumptions seem loose, $\phi = 1$ is still a natural choice as another value would require justifying why you think there is bias in the viscosity and how large you think that bias is. Regarding the choice for $C$, you are asking the quite a lot of the inverse solver if you supply a uniform initial guess as the field can vary by orders of magnitude. The assumption is made that under the shallow-stream approximation, most of the stress-balance on grounded ice is accounted for by sliding and gravity. In that case, a $C$ field that accounts for the grounded ice speed will be close to the $C$ field when the full inverse problem is performed for both control parameters over grounded and floating ice. I have changed the wording of this section slightly to make these points in the article. |
|---|---|---|
| 11 | Lines 194-7 (also Lines 226-30, 241-245): I think a summary of all experiments should be included at the end of the methods section, before going into the results. This will help to show readers exactly what you're doing in the context of methodology you've described. Introducing the exact cases during the results section seems a bit late. | This is a good point. I have attempted to make this clearer in the methods section by including lists of simulations for both snapshot and time-dependent problems. |
| 12 | Lines 203-4: The subpanel letters do not match the figure. These should be d,e,f not e,f,g | Thank you very much, I've fixed this now. |
| 13 | Lines 283-5: This is worded quite vaguely. If a reference to the previous paper is required (I would argue it is not here), be clear about what suggestion is being referred to. | I have changed this to read: "This suggests that observations of surface fracture on grounded ice have limited use in reducing the degeneracy associated with mixing between $C$ and $\phi$ fields" and removed the reference to a previous work by the authors. |
| 14 | Lines 334-6: The chosen value should also be labelled on Fig.5. In fact, it would be good to have the values labelled for each circle on the figure. | I have added a labels to each of the circles in figure 5 as suggested. |

**Responses to comments from Reviewer #2**

**Reviewer 2:** This study investigates the use of surface fracture and strain rate data in constraining inversions for ice rheology. The study considers two applications – the "snapshot" inversion infers both ice viscosity and basal friction in a single timepoint and the "time-dependent" inversion infers viscosity on an ice shelf at many points in time. The study finds that the inclusion of this additional information into regularization terms can alter the estimates found by the inverse method and possibly allows for an improved physical representation of ice viscosity in the inversion. The addition of this new data appears to be most useful on floating ice.

The application of more data, particularly that of surface fracture, to constrain glaciological inversions is a potentially very useful contribution, as inverse methods are widely used to initialize models and investigate drivers of ice sheet change. The study itself is very applicable to The Cryosphere. Below I describe some comments about the work itself and the presentation.

**Response:** We would like to thank the reviewer for their compliments on the content of the article and insightful review.

The study focuses on the application of these new methods to a case study of Pine Island Glacier. This makes it challenging to draw a concrete conclusion about whether this new data does improve the inversion because we don't know what the "right answer" is. Without knowing what ice softness is in Pine Island Glacier, it's hard to know how to compare these different cases the authors present (no regularization, heuristic regularization, data-informed regularization) rather than to say that they are different in certain ways. It seems to hamper the ability for the authors to suggest that one way is "better" than the other. One way of evaluating this is comparing the misfits to see if one regularization technique improves the optimization; however, in evaluations of Figures 2-4, there doesn't appear to be enough of a significant difference in the misfits to suggest that the data-informed regularization can produce more physical insight than the heuristic regularization. The authors are very careful and measured in the way they speak about these comparisons, which I think is a strength of this manuscript – they do acknowledge cases where the inclusion of this data does not appear to contribute to the inversion (e.g. on grounded ice). However, I still struggle with what the takeaways should be if there is such a difficulty in comparing between these cases. Possibly a clearer approach might be to test this technique on a synthetic case that approximates the PIG case study, in which a synthetic fracture field is imposed and a relationship between that fracture field and viscosity is assumed. This would provide a more straightforward way to compare between the cases presented in the manuscript and enhance the takeaways for the reader.

**Response:** This is a very good point, and I understand the desire to make more quantitative conclusions, however I think that attempting to do so might end up being detrimental to the study. Firstly, the ill-posedness of the problem means that we should not draw too many conclusions from the misfit. As the reviewer points out, an alternative

is to set up a problem in which the solution is essentially known a priori, e.g. synthetic examples. I did consider this when developing the work, however, I could not think of a way of doing so that would be unbiased. This would certainly have been the case had we imposed a relationship between the fracture field and viscosity in the set up of the experiments. It would also not be possible, for example, to make use of a fracture model, as there are no agreed-upon methods of doing so. Given that the effect of fractures on the rheology is always unknown prior to carrying out the inverse problem, I am still of the opinion that it is most appropriate to carry out these experiments on real data, and make-do with more qualitative statements about the success of the approach. It is my hope that there are important and interesting conclusions that people can draw from the article anyway. For example, the benefit of including fracture data in constraining damage/softness fields on floating ice is demonstrated convincingly, and the article provides a valuable demonstration that one can make use of additional data to change the solutions of the inverse problem.

**Responses to specific comments from Reviewer #2**

The description of the methods I found to be often hard to understand, in terms of the organization of the methods section and the wording of the explanations:

| | Reviewer 2 | |
|---|---|---|
| ID | Reviewer Comment | Response |
| 1 | A bit more explanation for how fractures are identified and how those fractures are converted into a continuous field to produce f would be helpful here, especially for those that haven't read the previous papers that describe these methods. | I have added a paragraph to the methods describing briefly how crevasse data are generated. I have also added a note that the smoothing is done by convolving the maps with a Gaussian kernel. |
| 2 | Line 44: the relationship between softness and stiffness seems to imply that stiffness is bounded between 0 and 1 – is this the case, and if so, why does this need to be the case? Stiffness appears to be simply a multiplicative factors on viscosity, in which case I don't see why viscosity can't vary by orders of magnitude | (For reference, this refers to the relation $\varphi = (1-\phi)$ where $\varphi$ represents softness and $\phi$ represents stiffness.) It is true that stiffness is bounded at the bottom by 0 (leading to an upper bound on softness of 1) which just prevents negative viscosities. In general, the stiffness is not bounded at the top by 1, meaning one can have negative softness. In our case, we do bound our stiffness at the top by 1, though it makes little difference to the solutions. It is right to say that the effective viscosities can vary by orders of magnitude, but this is almost all accounted for by Glen's flow law already. Where it is not, and we need the multiplicative scalar, important changes in viscosity are invariably on the side of reduced viscosity. We account for this with a stiffness that can reduce all the way to zero. |
| 3 | Lines 151-153 form the key description of the "snapshot" inversion and yet I found this to be challenging to understand. What is epsilon meant to represent physically? What is $\gamma$, physically? I also found it challenging to understand $\xi$ and its relationship with $\phi$. Having a clearer description of all these parameters would be very useful. | This is a good point, I have tried to make this clearer. Earlier on in the manuscript, I have included the line:
 "In essence, [$\xi$] should reflect our confidence in our initial guess for the ice rheology."
 I have then changed the line in question to read:
 "In the case of the snapshot inverse problem, the assumption we wish to encode in our prior for $\phi$ is that $\phi \sim \mathcal{N}(1, \gamma^2)$ where $\xi \to 1$, and $\gamma$ is a small number related to the strength of the prior."
 which removes one of the parameters in the original sentence ($\epsilon$) and states that $\gamma$ is related to the strength of our prior - which goes on to be related to the regularisation parameter. Hopefully this is clearer and will be readable by most - maybe with the use of Appendix A if required. |

| 4 | The L-curve section seems to be most applicable in the methods section, as I found myself wondering while reading how the regularization parameters were chosen and whether there was an L-curve-style approach to finding them. For example, lines 165-166 mention that there is an independent search for the regularization parameters but without further information it is hard to understand what this means. | The inclusion of this section is not so much to explain the method, but to make a more general point about the use of L-curve analysis when carrying out inverse problems. Rather than moving the section, I have included a sentence in the methods section explaining that L-curve analysis is used at each juncture to tune regularisation parameters. Hopefully, that means the section on L-curves in the discussion makes slightly more sense in the context of the rest of the article. |
|---|---|---|
| 5 | The term "high" shear strain rates is used often but not defined until line 145. A definition earlier (when it is first referenced) would be useful. | I tried moving the definition further up, but it seemed a little out of place. Instead, I have included a parenthetical "defined below". I can change this if the reviewer still thinks it is required. |
| 6 | Lines 128-130 imply that $\xi$ is a mask of only 0 and 1 values, but Figure 1 makes it seem like $\xi$ is continuous. | Good point. I have changed this to state that the field "goes to 0/1" rather than "is 0/1" in different areas. |
| 7 | Some of the equations (especially the regularization equations, such as Equations 10 and 11) could use much more explanation to describe what the terms mean and to remind the reader what the parameters are (I had trouble, for example, remembering the distinction between $f$ and $\xi$). | I have rewritten this section, including reducing the number of parameters one needs to keep track of and introducing $\xi$ earlier. I hope the various equations are now easier to follow. |
| 8 | The paragraph in lines 131-139 state that there are some things to note in the fracture data that are useful to understand the stress balance of PIG but the paragraph doesn't explain what the implications to the stress balance are. | Excellent point! I have added two sentences, one about the area of grounded crevasses, and one about crevassing in the shear margins: "If this is indeed an area in which membrane stresses form a significant component of the stress balance, the presence of crevasses deeper than the firn layer could have implications for the dynamics of this region by changing the horizontal transmission of stress." and: "Viscous deformation in shear margins can account for a significant portion of the stress budget of an ice shelf, so changes to the large-scale rheology in such locations will influence the distribution of stress throughout the ice shelf." |
| 9 | Line 200 – "The phi fields in each case are substantively different..." – it took me a while to understand what the different "cases" were (it is clear upon looking at the figure but it may be helpful to state this in the text as well) | Hopefully this is clearer in the modified manuscript. The 'cases' are outlined in the methods section and at the beginning of the results section. |
| 10 | Line 202 – "of even slow-flowing ice streams" – I wasn't sure what the "ice streams" were referencing here. | Good point. I have changed "ice streams" to "parts of the glacier". |

| 11 | Figures 2 and 4 – it would be helpful visually to add more labels to the colorbars rather than just the top and bottom labels. It could also be a useful diagnostic to visualize the misfit as a percentage of the observed velocity, to give some context to the absolute numbers. | Thank you for the comment. I have added more labels to the colourbars for all figures throughout the manuscript. Hopefully that makes things easier to interpret generally. I have not added the relative misfit as the important thing for these figures is the difference between the cases, rather than the misfits themselves. Adding an extra row to the figures makes them look a bit cluttered while not adding much. |

**Correspondence:** T. Surawy-Stepney (t.surawystepney@leeds.ac.uk)

**Abstract.** Numerical models used to simulate the evolution of the Antarctic Ice Sheet require the specification of basal boundary conditions on stress and local deviations in the assumed material properties of the ice. In general, scalar fields relevant to these unknown components of the system are found by solving an inverse problem given observations of model state variables - typically ice flow speed. However, these optimisation problems are ill posed, resulting in degenerate solutions and poor conditioning. In this study, we propose the use of fracture and strain rate data to provide prior information to the inverse problem, in an effort to better constrain the inferred ice softness compared to more heuristic regularisation techniques. We use Pine Island Glacier as a case study and consider both a snapshot inverse problem in which ice softness and basal slip parameters are sought simultaneously over the glacier as a whole, and a time-dependent problem in which ice softness alone is sought over the floating ice shelf at regular intervals. In the first case, we construct a prior encoding the assumption that the ice softness will be close to our initial guess except from where we see fractures or high shear strain rates in satellite data. We investigate the solutions and conditioning of this data-informed inverse problem versus alternatives. The second proposed method makes the assumption that changes to ice softness occurring on monthly-to-annual timescales will be dominated by the fracturing of ice. We show that these methods can result in softness fields on floating ice that visually mimic fracture patterns without significantly affecting the solution misfit, perhaps leading to greater confidence in the softness fields as a representation of the true material properties of the ice shelf.

**1 Introduction**

Large-scale ice sheet models commonly treat ice within the paradigm of continuum mechanics - as a shear thinning viscous fluid; an approach that has been successful in modelling the behaviour of large ice masses relatively cheaply (e.g. Seroussi et al. (2020)). Within this framework, the flow of the ice can be accounted for in large part by a balance between gravity, viscous stress due to internal deformation and frictional stress at ice/bedrock interfaces. To close the system and allow the model to solve for ice speed, equations relating viscous and frictional stresses to ice speed are specified, informed by laboratory data and physical arguments.

The former 'constitutive relation' very often takes the form of Glen's flow law:

$$\tau_{ij} = 2\eta\dot{\varepsilon}_{ij}, \qquad \text{where } \eta = \frac{1}{2}A(T)^{-\frac{1}{n}}\epsilon^{\frac{1}{n}-1} \tag{1}$$

where $\tau_{ij}$ is the deviatoric stress tensor, $\dot{\varepsilon}_{ij}$ is the strain rate tensor, $\epsilon$ is its second invariant, $\eta$ is the strain-rate-dependent effective ice viscosity  and $A(T)$ is a temperature-dependent rate factor. The value of the exponent $n$ is dependent on the particular mechanisms by which creep occurs within the ice and various properties of the crystal grains (e.g. Haefeli (1961)), and takes a value between 1 and  4 in most cases. (Here, we take the common reference value of $n = 3$.) It is possible to treat $A(T)$ and/or $n$ as free parameters that can be fitted to observations, given the uncertainties involved in both and the different physical mechanisms that distinguish them.  Frequently, however, these are prescribed *a priori* and a  stiffness field $\phi(x)$  is defined over the domain to account for unknown deviations in the expected ice rheology. As such, eq. (1) becomes $\tau_{ij} = 2\phi\eta\dot{\varepsilon}_{ij}$. Used in this way, $\phi$ approximates the effect of uncertainties in the temperature and thickness fields, regional changes in the temperature dependence of Glen's flow law, deviations from the assumed isotropy of creep deformation and, of particular interest to this study, fractures in the ice at different lengthscales. Often, a softness field $\varphi$ is defined in relation to the stiffness field by $\varphi = (1 - \phi)$.

The relation between frictional stress and basal sliding speed is known as a  sliding law, and has a functional form that depends on a number of often poorly constrained factors such as the expected amount of deformation of ice around topographic features in the bed, sliding over smooth bedrock, and shearing of the sub-glacial till. A single sliding law is often combined with a spatially varying  basal slip parameter $C(x)$ to approximate this stress:

$$\boldsymbol{\tau}_b = Cf(\boldsymbol{u}). \tag{2}$$

 Given a constititutive relation and sliding law defined as above, the equations

~~where $\boldsymbol{u} = (u_x, u_y)^{\top}$ is the horizontal velocity, $\bar{\eta}$ is the vertically-integrated effective ice viscosity, $\rho_i$ is the density of ice, $h$ is the ice thickness and $s$ is the ice surface. In this study we use a linear sliding law $f(\boldsymbol{u}) = \boldsymbol{u}$ for ease of computing adjoint sensitivities during the inverse problem. In this article, we also refer to the "softness" field $\varphi$ - related to the stiffness by $\varphi = (1 - \phi)$.~~

 solved by most large-scale ice sheet models contain a component dependent on $\phi$ (or a related scalar field performing an equivalent role) that represents viscous stress, a component dependent on $C$ that represents frictional stress, and a component representing gravitational driving. Therefore, for an ice sheet model to simulate real ice masses accurately,  these scalar fields must be well-constrained. In practice, they are typically inferred simultaneously from observations of ice speed using inverse methods - a suite of techniques for inferring model control parameters from observed state variables (MacAyeal, 1992) - (e.g. Petra et al. (2012); Arthern et al. (2015); Cornford et al. (2015); Gudmundsson
et al. (2019)).  Ice velocity data, rather than ice speed data, is also widely used in the community, and some methods of establishing current values for $C$ and $\phi$ also incorporate rates of thickness change into the inverse problem (e.g. Larour et al. (2014); Goldberg et al. (2015)) (though this relies on the model having an automatically differentiable forward solver). We don't explicitly consider these latter kinds of 'transient' inverse problem here, though the arguments we present still apply.

Regardless of its precise implementation, this inverse problem is ill-posed, resulting in solutions that are degenerate and highly dependent on noise in the input data (the problem, at least in its discrete form, is  *ill-conditioned*). To obtain reliable control fields, it is beneficial to replace this ill-posed problem  with a nearby well-posed one before solving it. The problem is sometimes simplified by solving for $C$ only on grounded ice, and $\phi$ on floating ice, thereby separating the two fields spatially and removing a portion of the degeneracy that arises from the mixing of  these fields (e.g. Goldberg et al. (2019)). However, though you would often expect $C$ to be the dominant control on grounded ice speed,  this may well not be true everywhere and an incorrect guess for $\phi$ could have consequences for transient simulations. Another approach  is to regularise the solution by providing additional constraints on the control fields. Such a regularised inverse problem takes the general form of the following optimisation problem:

$$(C,\phi) = \underset{C,\phi}{\mathrm{argmin}} \left\{ \mathcal{J}_m(u,u_o) + \alpha_C \mathcal{J}_C(C) + \alpha_\phi \mathcal{J}_\phi(\phi) \right\}, \quad \text{s.t.} \quad G(u,C,\phi) = 0 \tag{3}$$

where  $\mathcal{J}_m(u,u_o)$ is a misfit  functional calculating the distance of the  model output $u$ from the observed  data $u_o$ (often ice speed), $\mathcal{J}_C$ and $\mathcal{J}_\phi$ are regularisation  terms for the $C$ and $\phi$ fields, with strengths controlled by the parameters $\alpha_C$ and $\alpha_\phi$ respectively, and $G(u,C,\phi) = 0$ are the momentum balance equations  solved in the model's forward problem.

A popular approach, aimed at improving the conditioning of the problem by suppressing the amplification of high-frequency components of the input data, is to use Tikhonov regularisation in a form that favours either low spatial frequency or low amplitude components of the solution (e.g. Morlighem et al. (2013); Habermann et al. (2013); Brinkerhoff and Johnson (2013); Cornford et al. (2015)), e.g.:

$$\alpha_\phi \mathcal{J}_\phi(\phi) = \alpha_\phi \int_\Omega |\nabla \phi|^2 \, d^2\Omega. \tag{4}$$

However, this kind of regularisation is entirely heuristic and, when it comes to distinguishing $C$ and $\phi$, relies on assumed differences in the lengthscales over which changes in the control fields can influence strain rates. Generally, in regions without

85 significant shear, these lengthscales are not easily distinguished, and degeneracies between solutions for $C$ and $\phi$ proliferate. Additional difficulties arise when a control field contains distinct contributions with different spatial frequencies. For example, uncertainty in englacial temperature can vary on the scales of long-term atmospheric or geothermal heat sources, or over the width of a shear margin. Often, an imperfect but acceptable lengthscale is found by searching parameter space informed by heuristics such as L-curve analysis (Hansen and O'Leary, 1993; Hansen, 1994).

90

The aim of this study is to investigate whether the introduction of genuine prior information into the inverse problem  results in solutions that are more appealing than those found using other, heuristic regularisation methods.

95 Previous studies have investigated instances in which softness fields found through solving inverse problems have mirrored observed fracture features (Borstad et al., 2013; Surawy-Stepney et al., 2023a) - suggesting that the presence of fractures has the potential to dominate $\phi$. With recent advancements in observational methods for locating fractures in remote sensing data (Lai et al., 2020; Izeboud and Lhermitte, 2023; Zhao et al., 2022; Surawy-Stepney et al., 2023b), we are moving towards reliable data that can be used to inform us at least about this specific component of the softness field. Ranganathan et al. (2021)
100 showed previously that the use of strain rate data to weight the regularisation of $C$ and $\phi$ has the potential to reduce mixing between these control fields. The work presented here follows quite naturally from these results.

Here, we investigate two ways in which fracture and strain-rate observations can be used to inform the inverse problem to replace or complement existing heuristic methods. The first is to use
105  maps of surface fracture along with estimates of surface strain-rates to construct a prior distribution for $\phi$ for use in snapshot inverse problems (single optimisations carried out for a set of geometry and speed data collected at a specific instant in time).
110
 Next, we investigate the use of timeseries of fracture maps in constraining the solutions to inverse problems carried out over multiple timesteps on floating ice. We make the assumption that softness fields should vary on long timescales except from where we see changes to the pattern of fracture.
115 ~~of trust in the 3D temperature field we use. A more easily justified belief is that *changes* to $\phi$ on monthly-to-annual timescales are dominated by the fracturing of ice, as other contributions to $\phi$ are likely to vary on significantly longer timsescales. With this in mind, we initialise the inverse problem with heuristic regularisation, before imposing a regularisation that penalises the changes to $\phi$ except where we have seen the evolution of fractures in the observational data.this method~~these methods, that one can generate softness fields that mimic, in certain ways, the changing fracture patterns on the Pine Island Ice

120 Shelf between 2016 and 2021, without substantially affecting the solution misfit. This may have potential uses in constraining models that aim to evolve softness fields in response to englacial stresses.

**2 Methods**

The simulations presented in this article were performed using the BISICLES ice sheet model (Cornford et al., 2013). This is an adaptive mesh, finite volume model which we choose here to solve discretized versions of the two-dimensional shallow-stream

125 equations  :

$$\boldsymbol{\nabla} \cdot [\phi h \bar{\eta} (\boldsymbol{\nabla} \boldsymbol{u} + (\boldsymbol{\nabla} \boldsymbol{u})^\top + 2(\boldsymbol{\nabla} \cdot \boldsymbol{u})\mathcal{I})] - Cf(\boldsymbol{u}) - \rho_i g h \boldsymbol{\nabla} s = 0, \tag{5}$$

where $\boldsymbol{u} = (u_x, u_y)^\top$ is the horizontal velocity, $\bar{\eta}$ is the vertically-integrated effective ice viscosity, $\rho_i$ is the density of ice, $h$ is the ice thickness and $s$ is the ice surface. In this study we use a linear sliding law $f(\boldsymbol{u}) = \boldsymbol{u}$ for ease of computing adjoint sensitivities during the inverse problem.

130

Each inverse problem we consider in this article is of the form of eq. (3), with a misfit funtional of the form $\mathcal{J}_m(u, u_o) = \|u - u_o\|_2^2$. The inverse problems differ solely in the form of the regularisation terms $\mathcal{J}_\phi$. We solve each in BISICLES using a non-linear conjugate gradient method (Cornford et al., 2015).

135 Each simulation is carried out over Pine Island Glacier (PIG) in the Amundsen Sea Sector of West Antarctica with a domain encompassing the whole present-day drainage basin (Zwally et al., 2012). This region was chosen as it represents a potentially strong correspondence between fracturing and ice softness, given the abundant crevasses in the shear margins, upstream of the grounding line and the regular formation of rifts near the terminus, as well as the established dynamic impact of some of this fracturing (Joughin et al., 2021; Sun and Gudmundsson, 2023). Across the rest of Antarctica, we expect the link between

140 the dynamics of ice and the extent of fracturing to be weaker in general. We use a form of the rate factor $A(T)$ described in Cuffey and Paterson (2010), with an internal energy field generated using a $100\,000\,\mathrm{year}$ calculation in which surface temperature, thickness and velocity are held at present day values and the combined ice temperature and moisture fraction field $E = CT + Lw$ evolves toward equilibrium. We used a geometry defined by BedMachine-v3 (Morlighem, 2022), with  time-evolving calving front positions extracted from Sentinel-1 backscatter images. Each simulation used velocity

145 and fracture data from within a five-year period between November 2016 and November 2021. We used $200\,\mathrm{m}$ resolution, monthly-averaged ice velocity observations made using feature tracking applied to Sentinel-1 image pairs (Wuite et al., 2021) (https://cryoportal.enveo.at/data/) as the input data to the cost function and to estimate shear strain rates.

Crevasse data were generated according to the methods described in Surawy-Stepney et al. (2023b).

150 non-linear conjugate gradient method

(Cornford et al., 2015). This involves the application of deep-learning-based and other computer vision techniques to synthetic aperture radar (SAR) backscatter images from the Sentinel-1 satellite clusters, at 50 m spatial resolution. This produces maps showing the locations at which the surface expressions of crevasses and rifts are visible in the SAR data and include crevasses on floating and grounded ice. Of particular interest to this study are rifts on the Pine Island ice shelf, fractures in its shear margins, and the large field of grounded crevasses extending $\sim 100$ km upstream of the grounding line (Fig. 1 a). We use composite fracture maps that combine data from a month of SAR backscatter images, taking into account the differing visibility of crevasses imaged from different angles. The presence of obliquely overlapping Sentinel-1 frames is another reason for the choice of PIG as the location for this study.

**2.1 Fracture data assimilation in snapshot inverse problems**

The snapshot problem we consider is the joint estimation of $C$ and $\phi$ over Pine Island Glacier in May  2019 from mean ice speeds over the month.

[Figure]

**Figure 1.** Contributions to the field $\xi$, representing, in our prior for the softness field, where we have observations of surface fracture or high shear strain rates. a) SAR backscatter images over grounded and floating parts of Pine Island Glacier from May 2019 showing regions of visible crevassing: 1) surface crevasses on the grounded ice, 2) two almost-connected rifts near the Pine Island calving front, 3) the heavily 'damaged' southern shear margin of Pine Island Ice Shelf. b) The component of $\xi$ due to the observation of crevasse features, made from fracture maps developed in Surawy-Stepney et al. (2023b). Black boxes anticlockwise from the top show the locations of the SAR images a1, a2 and a3 respectively. c) The component of $\xi$ due to the presence of high shear strain rates. Background images to b and c are the MODIS Mosaic of Antarctica (Haran et al., 2021), and grounding lines (shown in black) are according to Rignot et al. (2016).

 The prior we construct for $\phi$  encodes the assumption that $\phi \approx 1$ away from regions of observed fracture or where there are high shear strain rates (which can contribute the effects of enhanced anisotropy, shear heating and microfracturing to $\phi$). In practise, this is equivalent to a form of Tikhonov regularisation using a diagonal Tikhonov matrix with entries weighted away from where we expect soft ice.

To construct this, we first form a field $\xi$ which  goes to 0 in regions which have high shear strain rates (defined below) or where fractures have been observed and to 1 elsewhere.  In essence, this should reflect our confidence in our initial guess for the ice rheology. We construct it as:

$$\xi = \min\{\xi_{\text{frac}}, \xi_{\text{shear}}\} \tag{6}$$

where $\xi_{\text{frac}}$ is low where we see fractures in satellite imagery (Fig. 1 b), and $\xi_{\text{shear}}$ is low where we see high strain rates (Fig. 1 c).

To construct $\xi_{\text{frac}}$, we first smooth the fracture map for May 2019, by convolving with a Gaussian kernel, to produce contiguous fracture fields on the grounded ice. We call  this fracture map $f$. Then $\xi_{\text{frac}} = 1 - f$ (Fig. 1 b). There are a few things to note in these fracture data of potential relevance to the stress-balance of the glacier. Firstly, we see a large contiguous area of surface fractures extending upstream from the grounding line and widening to cover a region in which previous studies have suggested membrane stresses are important in the stress-balance as basal stresses become small (Joughin et al., 2009) - something we see in our own solutions for basal stress. SAR images of this region show uniform coverage by closely-spaced surface fractures, almost identical in appearance (Fig. 1 a1). If this is indeed an area in which membrane stresses form a significant component of the stress balance, the presence of crevasses deeper than the firn layer could have implications for the dynamics of this region by changing the horizontal transmission of stress. Additionally, there is a rift (really, two rifts that are almost connected) near to the ice shelf terminus that led to the calving of a large tabular iceberg in February 2020 (Fig. 1 a2) - part of a series of calving events regarded to have had significant consequences for the dynamics of Pine Island Glacier (Joughin et al., 2021). Finally, there are a large number of fractures on the southern shear margin of Pine Island Ice Shelf (Fig. 1 a3). Viscous deformation in shear margins can account for a significant portion of the stress budget of an ice shelf, so changes to the large-scale rheology in such locations will influence the distribution of stress throughout the ice shelf.

We create $\xi_{\text{shear}}$, the strain-rate contribution to $\xi$, using the same velocity data that we use in our misfit  functional. To estimate the derivatives $\partial_i u_j$, we differentiated the velocity components using a method described in Chartrand (2017), using Tikhonov regularisation to promote smoothness (regularisation parameters were chosen with some trial-and-error, where preference was given to solutions in which regions of high shear varied smoothly over lengthscales comparable to the widths

of visible shear margins). Aligning the $x$-coordinate with local flow direction, we define regions of *high shear* to be those in which $|\dot{\varepsilon}_{xy}| > 0.1 \text{ a}^{-1}$. This threshold is a bit discretionary, though it corresponds to stresses within the range $90 - 320$ kPa of tensile strength suggested in Vaughan (1993) for a wide range of englacial temperatures. Then $\xi_{\text{shear}} = \max\{0, 1 - 10|\dot{\varepsilon}_{xy}|\}$ (Fig. 1 c) and $\xi = \min\{\xi_{\text{frac}}, \xi_{\text{shear}}\}$  (this looks like a combination of Fig. 1 b and c).

In the case of the snapshot inverse problem, the assumption we wish to encode  is that $\phi \sim \mathcal{N}(1, \gamma^2)$ whenever $\xi \to 1$, where  $\gamma$ is a small number  related to the strength of the prior. This can be written:

$$p_\Phi(\phi) \propto \exp\left(-\frac{1}{2\gamma^2}\int_\Omega (1-\phi)^2 \xi \, d\Omega\right). \tag{7}$$

Assuming the distribution of measurement errors is isotropic, with covariance $\sigma^2 \mathcal{I}$, this translates to a regularisation term:

$$\alpha_\phi = \frac{\sigma^2}{\gamma^2}, \quad \mathcal{J}_\phi(\phi) = \int_\Omega (1-\phi)^2 \xi \, d\Omega. \tag{8}$$

A greater exposition of this link between priors and regularisation parameters is given in appendix A.

 To understand how the introduction of prior information in the form of crevasse and strain-rate data changes the solutions to the inverse problem, we compare the solutions to those found using alternative regularisation methods. Hence, for the snapshot case, we perform three inverse problems over the full domain, starting with the same initial guesses for $C$ and $\phi$, with the same regularisation on $C$, with the following regularisation terms for $\phi$, defined in reference to eq. (3):

1. No regularisation: $\mathcal{J}_\phi(\phi) = 0$.

2. The widely-used heuristic regularisation: $\mathcal{J}_\phi(\phi) = \int_\Omega |\nabla\phi|^2 \, d\Omega$.

3. Our data-informed regularisation: $\mathcal{J}_\phi(\phi) = \int_\Omega (1-\phi)^2 \xi \, d\Omega$

The results are shown in section 3.1.

 We note that the initial guess for the control fields can have a large influence on the optimisation problem, as the closer it is to the desired solution, the more likely the optimisation to find that solution. For the $\phi$ field, we use an initial guess of 1 everywhere  (this is likely to be within an order of magnitude of the solution). The $C$  field can vary by orders of magnitude, so a uniform initial guess would be a poor choice. Instead, we take the view that the initial guess should be the field required to reproduce the observations on grounded ice as closely as possible with a uniform $\phi = 1$. This is reflective of

225  an assumption that grounded ice speed is largely accounted for by balance between gravity and friction (though we know this to be untrue). Hence, before carrying out the full optimisation including both control fields, we solve an inverse problem for $C$ with fixed $\phi = 1$, matching speeds only on grounded ice and use this as the initial guess for the joint inverse problem. This has the effect of  reducing the deviation of $\phi$ from 1 in the solution  and has the added bonus of allowing us to search independently for the regularisation parameters $\alpha_C$ and $\alpha_\phi$. In general, we carry out the search for regularisation

230  parameters using L-curve analysis (Hansen and O'Leary, 1993), though we consider this a heuristic to be used alongside other methods where necessary (section 4.3).

**2.2 Fracture data assimilation through time**

The use of fracture maps as a prior in the snapshot inverse problems makes an assumption about the relative contributions of

235  different uncertainties to $\phi$. For example, we have to have a certain amount of trust in the 3D temperature field we use. As previously noted,  $\phi$ also contains contributions from sources that cannot easily be distinguished by the spatial scales on which they vary. However, it seems likely that the contribution of fracturing to ice softness  varies on a shorter *temporal* scale than any other contribution. Hence, while attributing ice softness to the presence of fractures requires a large number of assumptions, we can reasonably attribute changes in ice softness

240   over monthly-to-annual timescales to the fracturing or

 healing of ice, and the advection of fractures. With this in mind, we consider the case of imposing a regularisation that penalises changes to $\phi$ in successive timesteps, except where we have seen the evolution of fractures in the observational data. Concretely, given a series of timesteps with times $\{t_i | i = 1, ..., n\}$, separated by $\Delta t$ (e.g. one month), we solve the following inverse problem for the control parameters $(C_i, \phi_i)$ at each timestep:

245  $$(C_i, \phi_i) = \underset{C_i, \phi_i}{\operatorname{argmin}} \{ \mathcal{J}_m(u_i, u_{o_i}) + \alpha_C \mathcal{J}_C(C_i) + \frac{\alpha_\phi}{\Delta t} \mathcal{J}_\phi(\phi_i) + \frac{\alpha_t}{\Delta t} \mathcal{J}_t(\phi_i, \phi_{i-1}) \},$$  (9)

 This is much the same as the snapshot inverse problem defined by eq. (3), though our regularisation term $\mathcal{J}_\phi(\phi_i, \phi_{i-1})$ now includes the softness fields in the current  and previous timesteps. Though not particularly sophisticated, a method such as described by Eq. (9) is immediately amenable to the introduction of fracture data  through its

250  inclusion in the regularisation term $\mathcal{J}_\phi$. Previous studies (Hogg et al., 2017; Selley et al., 2021) have used such a method with  $\mathcal{J}_\phi = \int_\Omega |\phi_i - \phi_{i-1}|^2 d\Omega$ and we modify this only slightly here. We propose the regularisation function:

$$\mathcal{J}_\phi = \int_\Omega (1 - |f_i - f_{i-1}|) \times |\phi_i - \phi_{i-1}|^2 d\Omega$$  (10)

where $f_i$ is the map showing the locations of fractures over the domain at time $t_i$. Hence, changes to the softness field are preferred in regions in which the fracture pattern has changed, with a strength that depends on the length of the timestep and the regularisation parameter $\alpha_t$.

We carry out such a procedure on Pine Island Glacier with 5 years of speed and fracture observations from December 2016 to December 2021, and timesteps of one month. This captures three calving events and the major disintegration of the southern shear margin of the ice shelf, and that of the calving front of Piglet Glacier (Joughin et al., 2021; Surawy-Stepney et al., 2023b). For each month, we use the mean speeds measured over that month as our observed speeds, and median fracture map composites.

We carry out two series of inverse problems, both starting with the same initial guess ($\phi$ field found using heuristic regularisation). One to act as a baseline, and the other reflecting our new approach:

[revised manuscript text omitted]

Goldberg, D. N., Gourmelen, N., Kimura, S., Millan, R., and Snow, K.: How Accurately Should We Model Ice Shelf Melt Rates?, Geophysical Research Letters, 46, 189–199, https://doi.org/https://doi.org/10.1029/2018GL080383, 2019.

Gudmundsson, G. H., Paolo, F. S., Adusumilli, S., and Fricker, H. A.: Instantaneous Antarctic ice sheet mass loss driven by thinning ice shelves, Geophysical Research Letters, 46, 13 903–13 909, https://doi.org/https://doi.org/10.1029/2019GL085027, 2019.

485 Habermann, M., Truffer, M., and Maxwell, D.: Changing basal conditions during the speed-up of Jakobshavn Isbræ, Greenland, The Cryosphere, 7, 1679–1692, https://doi.org/10.5194/tc-7-1679-2013, 2013.

Haefeli, R.: Contribution to the Movement and the form of Ice Sheets in the Arctic and Antarctic, Journal of Glaciology, 3, 1133–1151, https://doi.org/10.3189/S0022143000017548, 1961.

Hansen, P.: The L-curve and its use in the numerical treatment of inverse problems, in: InviteComputational Inverse Problems in Electrocar-
490 diology, WIT Press, inviteComputational Inverse Problems in Electrocardiology ; Conference date: 01-01-2000, 2000.

Hansen, P. C.: Regularization tools: A Matlab package for analysis and solution of discrete ill-posed problems, Numerical algorithms, 6, 1–35, https://doi.org/10.1007/BF02149761, 1994.

Hansen, P. C. and O'Leary, D. P.: The Use of the L-Curve in the Regularization of Discrete Ill-Posed Problems, SIAM Journal on Scientific Computing, 14, 1487–1503, https://doi.org/10.1137/0914086, 1993.

495 Haran, T. M., Bohlander, J., Scambos, T. A., Painter, T. H., and Fahnestock, M. A.: MODIS Mosaic of Antarctica 2003-2004 (MOA2004) Image Map, Version 2, https://doi.org/10.5067/68TBT0CGJSOJ, 2021.

Hogg, A. E., Shepherd, A., Cornford, S. L., Briggs, K. H., Gourmelen, N., Graham, J. A., Joughin, I., Mouginot, J., Nagler, T., Payne, A. J., Rignot, E., and Wuite, J.: Increased ice flow in Western Palmer Land linked to ocean melting, Geophysical Research Letters, 44, 4159–4167, https://doi.org/https://doi.org/10.1002/2016GL072110, 2017.

500 Izeboud, M. and Lhermitte, S.: Damage detection on antarctic ice shelves using the normalised radon transform, Remote Sensing of Environment, 284, 113 359, https://doi.org/10.1016/j.rse.2022.113359, 2023.

Joughin, I., Tulaczyk, S., Bamber, J. L., Blankenship, D., Holt, J. W., Scambos, T., and Vaughan, D. G.: Basal conditions for Pine Island and Thwaites Glaciers, West Antarctica, determined using satellite and airborne data, Journal of Glaciology, 55, 245–257, https://doi.org/10.3189/002214309788608705, 2009.

505 Joughin, I., Shapero, D., Smith, B., Dutrieux, P., and Barham, M.: Ice-shelf retreat drives recent Pine Island Glacier speedup, Science Advances, 7, eabg3080, https://doi.org/10.1126/sciadv.abg3080, 2021.

Lai, C.-Y., Kingslake, J., Wearing, M. G., Chen, P.-H. C., Gentine, P., Li, H., Spergel, J. J., and van Wessem, J. M.: Vulnerability of Antarctica's ice shelves to meltwater-driven fracture, Nature, 584, 574–578, https://doi.org/10.1038/s41586-020-2627-8, 2020.

Larour, E., Utke, J., Csatho, B., Schenk, A., Seroussi, H., Morlighem, M., Rignot, E., Schlegel, N., and Khazendar, A.: Inferred basal friction
510 and surface mass balance of the Northeast Greenland Ice Stream using data assimilation of ICESat (Ice Cloud and land Elevation Satellite) surface altimetry and ISSM (Ice Sheet System Model), The Cryosphere, 8, 2335–2351, https://doi.org/10.5194/tc-8-2335-2014, 2014.

[revised manuscript text omitted]

---

## Author Response (AR1)

**Responses to reviewer and editor comments for the article "Using observations of surface fracture to address ill-posed ice softness estimation over Pine Island Glacier"**

We would like to thank the editor and reviewers very much for the taking the time to read the article and for providing us with valuable and insightful feedback. (And also for their considerable patience in waiting for our responses.) All reviewer comments and responses are collated in this document, with each review reproduced in full and the editor's comments included at the bottom of the document. Responses to any general comments of the reviewers are coloured in teal, while responses to specific comments are tabulated afterwards.

A central theme of both reviews is that parts of the article should be restructured to make it easier to follow. To this end, I have made various structural changes, informed by specific comments, for example, moving text between the introduction and methods sections. I hope the reviewers find that these changes have improved the flow and clarity of the article. I have also added an additional figure (Fig. 3) that expands on the distribution of misfits for the three regularisation methods for the snapshot inverse problem. Finally, I have, without prompting from reviewers, removed the appendices after considering whether they were really necessary, though I am happy to add them back in.

**Responses to comments from Reviewer #1**

Reviewer 1: In this manuscript, the authors investigate the effect of assimilating more prior information into inversions for ice stiffness. The data informing the priors are strain rates, and locations of fracture derived from satellite imagery. Pine Island is chosen as a study area due to the large amount of fracturing observed there. Experiments are carried out using both snapshot and time-dependent inversion processes, using different regularisation. The results show that the use of this data in priors results in stiffness fields which better visually represent observed fracture patterns, without affecting the velocity misfit. The use of methods informed by fracture data could be important for improving inversions of floating ice, but is likely not have much impact on grounded areas. It is suggested that these methods would be best suited to diagnostic modelling and attempts to evolve stiffness fields through time.

This study will be valuable to a particular niche of ice flow modellers, and is certainly within the scope of The Cryosphere. I personally found it to be interesting, although I think wider interest will be limited as the focus is only on the inversion process and, by the authors' own admission, unlikely to be of much help to long-term predictive simulations.

My main issue with this manuscript is that it can be quite difficult to read, and is unclear at times. The introduction seems a little muddled, with some parts referencing specifics of this study among a more general review of the relevant issues. I would recommend moving anything specific to this study (sliding law, value of n in flow law etc.), and the more detailed discussion of reasoning behind the methods used found in the last paragraphs, into the methods section, so that it can all easily be found and doesn't over-complicate the introduction.

I also found the methods section difficult to follow in places. Section 2.1 would in my opinion benefit from being restructured. I also think the methods section should contain a clear summary of all the experiments which were carried out, as these are not all introduced until during the results section.

The scientific content of this manuscript is good, and worthy of publication, but I think work needs to be done to improve the clarity of its presentation. For this reason, I recommend publication after revisions.

Response: We would like to thank the reviewer for their positive comments about the article and their thorough and thoughtful review. I have restructured the introduction and methods sections (and the results to some degree) to make things clearer. Many of the changes made are in response to specific comments laid out in the review below.

**Responses to specific comments from Reviewer #1**

| Reviewer 1 |                                                                                                                                                                                                                                                                                                                                                                                                                                                                                                                                                                                           |                                                                                                                                                                                                                                                                                                                                                                            |
|------------|-------------------------------------------------------------------------------------------------------------------------------------------------------------------------------------------------------------------------------------------------------------------------------------------------------------------------------------------------------------------------------------------------------------------------------------------------------------------------------------------------------------------------------------------------------------------------------------------|----------------------------------------------------------------------------------------------------------------------------------------------------------------------------------------------------------------------------------------------------------------------------------------------------------------------------------------------------------------------------|
| ID         | Reviewer Comment                                                                                                                                                                                                                                                                                                                                                                                                                                                                                                                                                                          | Response                                                                                                                                                                                                                                                                                                                                                                   |
| 1          | Line 27: Is this a typo, or is the approximately equal sign there for a reason? If not a typo, please explain what is meant and be clear what value for n is being used in your work.                                                                                                                                                                                                                                                                                                                                                                                                     | This is not a typo, though I have made this less ambiguous by adding the sentence: "The value of the exponent $n$ is dependent on the particular mechanisms by which creep occurs within the ice and various properties of the crystal grains (e.g. Haefeli (1961)), and takes a value between 1 and 4 in most cases. Here, we take the common reference value of $n=3$ ". |
| 2          | Line 30: It may be helpful to write Eq.1 in a form which includes $\phi(x)$ for clarity.                                                                                                                                                                                                                                                                                                                                                                                                                                                                                                  | I have changed this sentence to be:
Here, we consider the approach in which these are
prescribed a priori and a 'stiffness' field $\phi(x)$ is
defined over the domain to account for unknown
deviations in the expected ice rheology, such that
eq. 1 becomes $\tau_{ij} = 2\phi\eta\dot{\epsilon}_{ij}$                                                   |
| 3          | Line 43: I think the sliding law used in this study
should be stated in the methods section rather than
the introduction                                                                                                                                                                                                                                                                                                                                                                                                                                                            | Thank you, I have moved this line into the methods.                                                                                                                                                                                                                                                                                                                        |
| 4          | Lines 48-51: It's a little unclear at points in this introduction whether you are talking about the specific process(es) used in your study, or more generally. As a more general point, some inversion processes use $u$ and $v$ velocity components as two separate observed fields, and some can also make use of thickness changes $dh/dt$ . This doesn't mean the problem is ever not ill-posed, but there is a greater variety in approaches that just using a single $u$ field. If this statement is referring to the specific process used in this study, please make this clear. | I hope that the changes made to the introduction and methods have addressed the issue of clarity and distinguished statements that relate to methods in general and those we use ourselves in the article. I have also added a couple of sentences that make it clear that other data can be included in the inverse problem as suggested.                                 |
| 5          | Lines 92-5: This detail probably belongs in the methods section                                                                                                                                                                                                                                                                                                                                                                                                                                                                                                                           | This has been moved into the methods.                                                                                                                                                                                                                                                                                                                                      |
| 6          | Lines 97-102: As above, better to put the detail in methods.                                                                                                                                                                                                                                                                                                                                                                                                                                                                                                                              | This has also been moved into the methods                                                                                                                                                                                                                                                                                                                                  |

| 7 | Lines 113-4: Could this point about the link between dynamics and fracturing over the rest of Antarctica be expanded on in the discussion?                                                                                                                                                                                                                                                                   | I appreciate the desire to expand on this, but I think it might be difficult to do so without moving into speculation. PIG has shown a strong connection between fracturing and broader dynamics over the last decade, e.g. the cited studies showing links between calving, shear margin degradation and changes in ice speed; also the very low basal stresses found far inland of the grounding line. There is a feeling that this is not replicated in other places round Antarctica, though there is actually little concrete evidence of that. For example, when carrying out inverse problems, I have not seen very low basal stresses on grounded ice in many other places, but I haven't actually done or seen proper analysis on it. I think it might be better to keep this to a short statement reflecting a generally held belief than include too much pontification. I could be persuaded otherwise though. |
|---|--------------------------------------------------------------------------------------------------------------------------------------------------------------------------------------------------------------------------------------------------------------------------------------------------------------------------------------------------------------------------------------------------------------|----------------------------------------------------------------------------------------------------------------------------------------------------------------------------------------------------------------------------------------------------------------------------------------------------------------------------------------------------------------------------------------------------------------------------------------------------------------------------------------------------------------------------------------------------------------------------------------------------------------------------------------------------------------------------------------------------------------------------------------------------------------------------------------------------------------------------------------------------------------------------------------------------------------------------|
| 8 | Lines 121-2: You refer to this past paper a few
times without detail. As it relates to an important
source of data in this study, a brief description of
the method would be useful in this section, or at
least mentioning that it uses a machine learning
technique to identify crevasses.                                                                                                  | I have included a short paragraph with a little more detail about this dataset.                                                                                                                                                                                                                                                                                                                                                                                                                                                                                                                                                                                                                                                                                                                                                                                                                                            |
| 9 | Lines 128-49: In my opinion, these paragraphs would benefit from a little restructuring. I think the definition of $\xi = \min\{\xi_{\text{frac}}, \xi_{\text{shear}}\}$ should be introduced first, defining what the components are, before then presenting the details of how the components are calculated. This would have made it easier for me to follow, although that may be a personal preference. | Thank you for the good suggestion, I have changed the structure as suggested.                                                                                                                                                                                                                                                                                                                                                                                                                                                                                                                                                                                                                                                                                                                                                                                                                                              |

| 10 | Lines 161-3: Could you give a reason for the choice of initial guesses? After stating that this can have a large influence on the optimisation, I feel a justification of the choice is required. Why not, for example, use a uniform guess for $C$ or a value of 0.5 for $\phi$ ?                                                                    | This is a very good point! The optimisation problem will be more easily solved if the initial guess is close to the optimal solution. If we think Glen's flow law with a choice of $n=3$ is correct, the ice is unbroken, and the temperature field we get from the thermomechanical spin-up described in the text is a good approximation, then taking $\phi=1$ is the right choice. Even if those assumptions seem loose, $\phi=1$ is still a natural choice as another value would require justifying why you think there is bias in the viscosity and how large you think that bias is. Regarding the choice for $C$ , you are asking the quite a lot of the inverse solver if you supply a uniform initial guess as the field can vary by orders of magnitude. The assumption is made that under the shallow-stream approximation, most of the stress-balance on grounded ice is accounted for by sliding and gravity. In that case, a $C$ field that accounts for the grounded ice speed will be close to the $C$ field when the full inverse problem is performed for both control parameters over grounded and floating ice.  I have changed the wording of this section slightly to make these points in the article. |
|----|-------------------------------------------------------------------------------------------------------------------------------------------------------------------------------------------------------------------------------------------------------------------------------------------------------------------------------------------------------|--------------------------------------------------------------------------------------------------------------------------------------------------------------------------------------------------------------------------------------------------------------------------------------------------------------------------------------------------------------------------------------------------------------------------------------------------------------------------------------------------------------------------------------------------------------------------------------------------------------------------------------------------------------------------------------------------------------------------------------------------------------------------------------------------------------------------------------------------------------------------------------------------------------------------------------------------------------------------------------------------------------------------------------------------------------------------------------------------------------------------------------------------------------------------------------------------------------------------------|
| 11 | Lines 194-7 (also Lines 226-30, 241-245): I think a summary of all experiments should be included at the end of the methods section, before going into the results. This will help to show readers exactly what you're doing in the context of methodology you've described. Introducing the exact cases during the results section seems a bit late. | This is a good point. I have attempted to make this clearer in the methods section by including lists of simulations for both snapshot and time-dependent problems.                                                                                                                                                                                                                                                                                                                                                                                                                                                                                                                                                                                                                                                                                                                                                                                                                                                                                                                                                                                                                                                            |
| 12 | Lines 203-4: The subpanel letters do not match the figure. These should be d,e,f not e,f,g                                                                                                                                                                                                                                                            | Thank you very much, I've fixed this now.                                                                                                                                                                                                                                                                                                                                                                                                                                                                                                                                                                                                                                                                                                                                                                                                                                                                                                                                                                                                                                                                                                                                                                                      |
| 13 | Lines 283-5: This is worded quite vaguely. If a reference to the previous paper is required (I would argue it is not here), be clear about what suggestion is being referred to.                                                                                                                                                                      | I have changed this to read: "This suggests that observations of surface fracture on grounded ice have limited use in reducing the degeneracy associated with mixing between $C$ and $\phi$ fields" and removed the reference to a previous work by the authors.                                                                                                                                                                                                                                                                                                                                                                                                                                                                                                                                                                                                                                                                                                                                                                                                                                                                                                                                                               |
| 14 | Lines 334-6: The chosen value should also be labelled on Fig.5. In fact, it would be good to have the values labelled for each circle on the figure.                                                                                                                                                                                                  | I have added a labels to each of the circles in figure 5 as suggested.                                                                                                                                                                                                                                                                                                                                                                                                                                                                                                                                                                                                                                                                                                                                                                                                                                                                                                                                                                                                                                                                                                                                                         |

**Responses to comments from Reviewer #2**

Reviewer 2: This study investigates the use of surface fracture and strain rate data in constraining inversions for ice rheology. The study considers two applications – the "snapshot" inversion infers both ice viscosity and basal friction in a single timepoint and the "time-dependent" inversion infers viscosity on an ice shelf at many points in time. The study finds that the inclusion of this additional information into regularization terms can alter the estimates found by the inverse method and possibly allows for an improved physical representation of ice viscosity in the inversion. The addition of this new data appears to be most useful on floating ice.

The application of more data, particularly that of surface fracture, to constrain glaciological inversions is a potentially very useful contribution, as inverse methods are widely used to initialize models and investigate drivers of ice sheet change. The study itself is very applicable to The Cryosphere. Below I describe some comments about the work itself and the presentation.

**Response:** We would like to thank the reviewer for their compliments on the content of the article and insightful review.

The study focuses on the application of these new methods to a case study of Pine Island Glacier. This makes it challenging to draw a concrete conclusion about whether this new data does improve the inversion because we don't know what the "right answer" is. Without knowing what ice softness is in Pine Island Glacier, it's hard to know how to compare these different cases the authors present (no regularization, heuristic regularization, data-informed regularization) rather than to say that they are different in certain ways. It seems to hamper the ability for the authors to suggest that one way is "better" than the other. One way of evaluating this is comparing the misfits to see if one regularization technique improves the optimization; however, in evaluations of Figures 2-4, there doesn't appear to be enough of a significant difference in the misfits to suggest that the data-informed regularization can produce more physical insight than the heuristic regularization. The authors are very careful and measured in the way they speak about these comparisons, which I think is a strength of this manuscript – they do acknowledge cases where the inclusion of this data does not appear to contribute to the inversion (e.g. on grounded ice). However, I still struggle with what the takeaways should be if there is such a difficulty in comparing between these cases. Possibly a clearer approach might be to test this technique on a synthetic case that approximates the PIG case study, in which a synthetic fracture field is imposed and a relationship between that fracture field and viscosity is assumed. This would provide a more straightforward way to compare between the cases presented in the manuscript and enhance the takeaways for the reader.

**Response:** This is a very good point, and I understand the desire for more concrete conclusions, however I think that attempting to do so might end up being detrimental to the study. The ill-posedness of the problem means that we are not able to draw too many conclusions about the solution from the misfit. As the reviewer points out, an alternative

is to set up a problem in which the solution is essentially known a priori, e.g. synthetic examples. I did consider this when developing the work, however, I could not think of a way of doing so that would allow us to generalise to real world data given the number of assumptions we would have to make (e.g., as the reviewer suggests, a relationship between crevasse field and softness). Given that the effect of fractures on the rheology is always unknown prior to carrying out the inverse problem, we are of the opinion that it is most appropriate to carry out these experiments on real data, and make-do with more qualitative statements about the success of the approach. It is my hope that there are important and interesting conclusions that people can draw from the article anyway. For example, the benefit of including fracture data in constraining damage/softness fields on floating ice is demonstrated convincingly, and the article provides a valuable demonstration that one can make use of additional data to change the solutions of the inverse problem.

I have added a little bit to the discussion explaining our approach (section 4.4 "Next steps"). I have also added another figure showing the distribution of misfits for the snapshot inverse problem for the three cases, which could be of interest.

(See also my response to the editor's comments, at the end of this document.)

**Responses to specific comments from Reviewer #2**

The description of the methods I found to be often hard to understand, in terms of the organization of the methods section and the wording of the explanations:

| Reviewer 2 |                                                                                                                                                                                                                                                                                                                                                                      |                                                                                                                                                                                                                                                                                                                                                                                                                                                                                                                                                                                                                                                                                                                                                                                                                                                                         |
|------------|----------------------------------------------------------------------------------------------------------------------------------------------------------------------------------------------------------------------------------------------------------------------------------------------------------------------------------------------------------------------|-------------------------------------------------------------------------------------------------------------------------------------------------------------------------------------------------------------------------------------------------------------------------------------------------------------------------------------------------------------------------------------------------------------------------------------------------------------------------------------------------------------------------------------------------------------------------------------------------------------------------------------------------------------------------------------------------------------------------------------------------------------------------------------------------------------------------------------------------------------------------|
| ID         | Reviewer Comment                                                                                                                                                                                                                                                                                                                                                     | Response                                                                                                                                                                                                                                                                                                                                                                                                                                                                                                                                                                                                                                                                                                                                                                                                                                                                |
| 1          | A bit more explanation for how fractures are identified and how those fractures are converted into a continuous field to produce f would be helpful here, especially for those that haven't read the previous papers that describe these methods.                                                                                                                    | I have added a paragraph to the methods describing
briefly how crevasse data are generated. I have
also added a note that the smoothing is done by
convolving the maps with a Gaussian kernel.                                                                                                                                                                                                                                                                                                                                                                                                                                                                                                                                                                                                                                                                 |
| 2          | Line 44: the relationship between softness and stiffness seems to imply that stiffness is bounded between 0 and 1 – is this the case, and if so, why does this need to be the case? Stiffness appears to be simply a multiplicative factors on viscosity, in which case I don't see why viscosity can't vary by orders of magnitude                                  | (For reference, this refers to the relation $\varphi=(1-\phi)$ where $\varphi$ represents softness and $\phi$ represents stiffness.) It is true that stiffness is bounded at the bottom by 0 (leading to an upper bound on softness of 1) which just prevents negative viscosities. In general, the stiffness is not bounded at the top by 1, meaning one can have negative softness. In our case, we do bound our stiffness at the top by 1, though it makes little difference to the solutions. It is right to say that the effective viscosities can vary by orders of magnitude, but this is almost all accounted for by Glen's flow law already. Where it is not, and we need the multiplicative scalar, important changes in viscosity are invariably on the side of reduced viscosity. We account for this with a stiffness that can reduce all the way to zero. |
| 3          | Lines 151-153 form the key description of the "snapshot" inversion and yet I found this to be challenging to understand. What is epsilon meant to represent physically? What is $\gamma$ , physically? I also found it challenging to understand $\xi$ and its relationship with $\phi$ . Having a clearer description of all these parameters would be very useful. | This is a good point, I have tried to make this clearer. Earlier on in the manuscript, I have included the line: "In essence, $[\xi]$ should reflect our confidence in our initial guess for the ice rheology." I have then changed the line in question to read: "In the case of the snapshot inverse problem, the assumption we wish to encode in our prior for $\phi$ is that $\phi \sim \mathcal{N}(1, \gamma^2)$ where $\xi \to 1$ , and $\gamma$ is a small number related to the strength of the prior." which removes one of the parameters in the original sentence $(\epsilon)$ and states that $\gamma$ is related to the strength of our prior - which goes on to be related to the regularisation parameter. Hopefully this is clearer and will be readable by most - maybe with the use of Appendix A if required.                                        |

| 4  | The L-curve section seems to be most applicable in the methods section, as I found myself wondering while reading how the regularization parameters were chosen and whether there was an L-curve-style approach to finding them. For example, lines 165-166 mention that there is an independent search for the regularization parameters but without further information it is hard to understand what this means. | The inclusion of this section is not so much to explain the method, but to make a more general point about the use of L-curve analysis when carrying out inverse problems. Rather than moving the section, I have included a sentence in the methods section explaining that L-curve analysis is used at each juncture to tune regularisation parameters. Hopefully, that means the section on L-curves in the discussion makes slightly more sense in the context of the rest of the article.                                                                                                                                                                                      |
|----|---------------------------------------------------------------------------------------------------------------------------------------------------------------------------------------------------------------------------------------------------------------------------------------------------------------------------------------------------------------------------------------------------------------------|-------------------------------------------------------------------------------------------------------------------------------------------------------------------------------------------------------------------------------------------------------------------------------------------------------------------------------------------------------------------------------------------------------------------------------------------------------------------------------------------------------------------------------------------------------------------------------------------------------------------------------------------------------------------------------------|
| 5  | The term "high" shear strain rates is used often but not defined until line 145. A definition earlier (when it is first referenced) would be useful.                                                                                                                                                                                                                                                                | I tried moving the definition further up, but it seemed a little out of place. Instead, I have included a parenthetical "defined below". I can change this if the reviewer still thinks it is required.                                                                                                                                                                                                                                                                                                                                                                                                                                                                             |
| 6  | Lines 128-130 imply that $\xi$ is a mask of only 0 and 1 values, but Figure 1 makes it seem like $\xi$ is continuous.                                                                                                                                                                                                                                                                                               | Good point. I have changed this to state that the field "goes to $0/1$ " rather than "is $0/1$ " in different areas.                                                                                                                                                                                                                                                                                                                                                                                                                                                                                                                                                                |
| 7  | Some of the equations (especially the regularization equations, such as Equations 10 and 11) could use much more explanation to describe what the terms mean and to remind the reader what the parameters are (I had trouble, for example, remembering the distinction between $f$ and $\xi$ ).                                                                                                                     | I have rewritten this section, including reducing the number of parameters one needs to keep track of and introducing $\xi$ earlier. I hope the various equations are now easier to follow.                                                                                                                                                                                                                                                                                                                                                                                                                                                                                         |
| 8  | The paragraph in lines 131-139 state that there are some things to note in the fracture data that are useful to understand the stress balance of PIG but the paragraph doesn't explain what the implications to the stress balance are.                                                                                                                                                                             | This is an excellent point. I have added two sentences, one about the area of grounded crevasses, and one about crevassing in the shear margins:  "If this is indeed an area in which membrane stresses form a significant component of the stress balance, the presence of crevasses deeper than the firn layer could have implications for the dynamics of this region by changing the horizontal transmission of stress."  and:  "Viscous deformation in shear margins can account for a significant portion of the stress budget of an ice shelf, so changes to the large-scale rheology in such locations will influence the distribution of stress throughout the ice shelf." |
| 9  | Line 200 – "The phi fields in each case are substantively different" – it took me a while to understand what the different "cases" were (it is clear upon looking at the figure but it may be helpful to state this in the text as well)                                                                                                                                                                            | Hopefully this is clearer in the modified manuscript. The 'cases' are outlined in the methods section and at the beginning of the results section.                                                                                                                                                                                                                                                                                                                                                                                                                                                                                                                                  |
| 10 | Line $202$ – "of even slow-flowing ice streams" – I wasn't sure what the "ice streams" were referencing here.                                                                                                                                                                                                                                                                                                       | Thank you, that was unclear. I have changed "ice streams" to "parts of the glacier".                                                                                                                                                                                                                                                                                                                                                                                                                                                                                                                                                                                                |

Figures 2 and 4 – it would be helpful visually to add more labels to the colorbars rather than just the top and bottom labels. It could also be a useful diagnostic to visualize the misfit as a percentage of the observed velocity, to give some context to the absolute numbers.

Thank you for the comment. I have added more labels to the colourbars for all figures throughout the manuscript. Hopefully that makes things easier to interpret generally. I have not added the relative misfit as the important thing for these figures is the difference between the cases, rather than the misfits themselves. Adding an extra row to the figures makes them look a bit cluttered while not adding much.

**Responses to comments from the editor**

Your manuscript has received two constructive reviews. Both reviewers have acknowledged the scientific quality of your work. Thank you for addressing their comments thoughtfully and for incorporating improvements into the revised version of your manuscript. However, I have identified several concerns that I believe need to be addressed in your next revision:

- 1: Consistent with the feedback from Reviewer #2, my primary concern is "whether this new data does improve the inversion". I understand the challenge to draw conclusions with synthetic data. While I leave the ultimate decision in your hands, I strongly recommend that you carefully reconsider Reviewer #2's suggestions, particularly regarding the inclusion of additional experiments. If you choose not to conduct these experiments, I advise adding a detailed discussion section that explicitly explains your rationale. This should also outline what you believe would constitute reasonable next steps for anyone aiming to build on your work.
- 2: Quantification of results: As a manuscript focused on numerical modeling, I find it surprising that the results section contains minimal quantitative metrics, especially when comparing with observations. Specifically, I encourage you to include quantitative measures such as the min, max, and mean misfits to assess model performance more comprehensively. Such metrics would be particularly useful for interpreting the results presented in Figures 2 (d-f) and Figures 3 (d-f). Providing these objective metrics will not only help readers better evaluate model performance but may also offer opportunities to enhance the depth of your discussion.

Therefore, I now strongly encourage you to submit a revised version that addresses these remarks. Based on this revised manuscript, I will invite both reviewers to assess whether the improvements warrant acceptance for publication in The Cryosphere.

Response: Thank you very much again for considering the revised article and for your constructive comments. I hope that you find the additions to the article and responses below satisfactory.

1: The question of "whether this new data does improve the inversion" is a bit difficult to answer exactly. Ultimately, the best approach is the one that most often gives you the right answer. However, it is difficult to judge what the right answer is without knowing it a priori. There are problems for which a good misfit indicates a good solution so, by performing different experiments and analysing the misfits, one can understand the quality of the method. However, it is fundamental to the nature of ill-conditioned problems (such as this, and others that naturally arise in viscous flow problems) that this is not true. Hence, though we could analyse the misfit much more extensively in the article, we have no guarantees that it would tell us anything reliable about the quality of the solutions. (Having said that, I have added a bit more about the misfit, described below in my response to your second suggestion.)

As you point out, an alternative approach is to use synthetic experiments in which the solution is known a priori. An example we could have set up is to have determined a steady-state softness field by prescribing a crevasse-depth formula and interpreting the relative depth of crevasses as 'damage' (1-softness). Then used the location of the crevasses to construct our prior, then recover the solution and show it works better than other regularisation methods (and you could even probably state by how much the new approach is better). This is appealing in a number of ways, not least because it would have been more straightforward than using real data. However, due to the large number of assumptions we can't justify and whose effect we can't quantify, we judged that experiment like that would not tell us anything useful about the real-world case. We are still of the opinion that a better approach is to consider a real-life example, and look for features in the solutions we would expect to see. In so doing, we are sacrificing the ability to make quantitative statements for the assurance that we are at least measuring something sensible.

I have added a short section to the discussion (Section 4.4 "Next Steps") explaining this rationale. I think the section is redundant, but am happy to keep it if the reviewers agree that it improves the article.

There is also the question that your conception of an improved method for the inverse problem will probably depend on what you want the solution for. We discuss this in the article. It might be that, even if the softness field you recovered were a perfectly faithful representation of the bulk material properties of the ice at that moment in time, it might be less useful to you than one that is not. (E.g. if you want a generic softness field that isn't too wrong for many timesteps of a transient simulation).

Overall, this is probably not going to be the kind of article that someone can easily skim to find a metric telling them the best approach to employ in their model. But we (and, I think, the reviewers) are of the opinion that it is still of value and interest to the community - perhaps more to those who have a particular interest in inverse problems. I think that makes it worthy of publication in The Cryosphere.

2: Thank you for this suggestion, I agree that there could be more numbers in the article. (Though, related to the first point, it would be a shame if these quantitative statements about the misfit overshadowed the much more important qualitative discussion of the solutions.) I have added a figure to the article (Fig. 3) that shows the distributions of misfits on grounded and floating ice for the snapshot inverse problem (there is already a subfigure looking at the misfit for the time-dependent problem) and various statements about the mean misfits for different cases. I think this will be interesting to the reader and hopefully satisfies the requirement for including more numbers.

**References**

Haefeli, R.: Contribution to the Movement and the form of Ice Sheets in the Arctic and Antarctic, Journal of Glaciology, 3, 1133–1151, https://doi.org/10.3189/S0022143000017548, 1961.

---

## Referee Report (RR1)

Review for revised version of "Using observations of surface fracture to address ill-posed ice softness estimation over Pine Island Glacier" by Surawy-Stepney et al.

Thanks to the authors for the effort taken to thoroughly and thoughtfully address the comments made by myself and the other reviewer. The scientific content is as good as before, and the manuscript now appears better organised and easier to follow. I think it is very interesting work and entirely appropriate to The Cryosphere. I am satisfied with the changes and happy to recommend publication.

As picked up on by the other reviewer and the editor, the results in this paper are more qualitative than quantitative, but in this case I don't think that's a weakness. I am of the opinion that outputs from inversion processes are often better judged by looking at the distribution of output fields (such as those shown in Fig. 5) than by quantitative measures (as long as the misfit is reasonable, which it seems to be here). The authors are careful to point out the limitations of the methodology and do not overstate their findings, giving a realistic view of when the method would be most usefully applied.

The addition of the 'Next Steps' section in the discussion to explicitly explain the authors' reasoning should aid readers in understanding decisions made in devising this work, and opens the way for other interested modellers to explore the idea further with synthetic experiments. However, it does also emphasise a remaining small complaint I have with the manuscript. The authors refer a few times to what they would 'expect' to see (e.g. lines 118, 352, 385) but without explicitly stating *why* they would expect that. The manuscript is fine as it is, but these points would be stronger if the reasoning behind these expectations were made clear, and I think the authors should consider this.

I will make one further comment, although this is not something which would necessarily need changing. It's down to personal preference and I mention it for the authors to consider at their discretion. When presenting misfit as in Fig. 2, I find it more useful to either present the misfit as a proportion of the measurements, or to see the measurements plotted alongside. In this case I know that the speed of Pine Island means that maximum misfit on the order of 200-300 m/a is not unreasonable, but others may not be so familiar with the glacier.

Below I list some minor typographical errors I noticed while reading through, which the authors may wish to correct:

Line 42 - "constitutive"

Line 110 – "functional"

Line 240 - "snapshot"

Fig 2 & Fig 3 – Here you use the American "regularization", while you use the British "regularisation" in the text

Line 287 – "from" is repeated

Line 348 – "principal"

---

## Author Response (AR2)

**Responses to second set of reviewer comments for the article "Using observations of surface fracture to address ill-posed ice softness estimation over Pine Island Glacier"**

We would like to thank the reviewers and editor once again for their very fair and thoughtful assessments of the article, and for their patience in waiting for us to address their comments. The reviews comments are reproduced here and my responses are coloured in teal.

Having previously made the case that it might not be hugely valuable to include any synthetic experiments in the manuscript, we have now decided to include a section that uses a synthetic ice stream to justify the basic premise of the method (in the snapshot case) and set up some expectations for what the method might do (Section 3: A synthetic example). This sits between the methods and main results sections. This should provide some additional insight to the reader about the theoretical efficacy of the method, but stops short of making claims we would struggle to fairly translate to the real-world setting. We think that this addresses the main remaining concerns of the reviewers (and corresponds to the first option for doing so suggested by the editor in their latest report).

**Responses to comments from Reviewer #1**

Reviewer 1: Thanks to the authors for the effort taken to thoroughly and thoughtfully address the comments made by myself and the other reviewer. The scientific content is as good as before, and the manuscript now appears better organised and easier to follow. I think it is very interesting work and entirely appropriate to The Cryosphere. I am satisfied with the changes and happy to recommend publication.

As picked up on by the other reviewer and the editor, the results in this paper are more qualitative than quantitative, but in this case I don't think that's a weakness. I am of the opinion that outputs from inversion processes are often better judged by looking at the distribution of output fields (such as those shown in Fig. 5) than by quantitative measures (as long as the misfit is reasonable, which it seems to be here). The authors are careful to point out the limitations of the methodology and do not overstate their findings, giving a realistic view of when the method would be most usefully applied.

The addition of the 'Next Steps' section in the discussion to explicitly explain the authors' reasoning should aid readers in understanding decisions made in devising this work, and opens the way for other interested modellers to explore the idea further with synthetic experiments. However, it does also emphasise a remaining small complaint I have with the manuscript. The authors refer a few times to what they would 'expect' to see (e.g. lines 118, 352, 385) but without explicitly stating why they would expect that. The manuscript is fine as it is, but these points would be stronger if the reasoning behind these expectations were made clear, and I think the authors should consider this.

Thank you for the very thoughtful comment. I have added a note accompanying the use of 'expect' in line 118 that lets readers know that this is due to the coincident changes in dynamics and structural integrity on the glacier. As for lines 352 and 385 (and less explicitly elsewhere) hopefully the new section "A synthetic example" goes some way to informing the reader of our expectations.

I will make one further comment, although this is not something which would necessarily need changing. It's down to personal preference and I mention it for the authors to consider at their discretion. When presenting misfit as in Fig. 2, I find it more useful to either present the misfit as a proportion of the measurements, or to see the measurements plotted alongside. In this case I know that the speed of Pine Island means that maximum misfit on the order of 200-300 m/a is not unreasonable, but others may not be so familiar with the glacier.

This is a very good point. I have added a note to the end of section 4.1 describing the speeds we see on Pine Island Glacier which should help put these misfits in context.

Below I list some minor typographical errors I noticed while reading through, which the authors may wish to correct:

• Line 42 – "constitutive"

- $\bullet \ \, \mbox{Line } 110-\,\mbox{"functional"}$
- $\bullet$  Line 240 "snapshot"
- $\bullet$  Fig 2 & Fig 3 Here you use the American "regularization", while you use the British 'regularisation" in the text
- $\bullet$  Line 287 "from" is repeated
- Line 348 "principal"

Thank you for your keen eye! I have corrected these.

**Responses to comments from Reviewer #2**

**Reviewer 2:** Thanks to the authors for addressing and replying to all of my earlier comments. The changes have greatly improved the clarity of the manuscript.

My only remaining comment has to do with the takeaways of the paper. I mentioned in my initial review that I found it challenging to draw any concrete conclusions about the results presented given that there is no known "right answer" to the problem, and therefore no metrics for determining whether the inclusion of fractures as prior information improved the results of the inversion.

This concern still stands to some degree. I acknowledge the point that there are many assumptions that would have to be imposed in order to construct any synthetic experiments that would quantitatively address the question (e.g. the effect of fractures on ice softness). That being said, I do think there are relatively well-known approaches to these that the authors could take. For example, the effect of fractures on ice softness is parameterized in largely the same way in every continuum damage mechanics study (see, for example, Chris Borstad's studies), and there would be a reasonably strong justification for following these approaches. The imposition of a crevasse field and choices about other processes affecting ice softness would need to be assumed as well but I do think these are justifiable choices that can also be evaluated in a systematic way (e.g. imposing many different kinds of crevasse fields, doing multiple experiments with different processes affecting ice softness). I believe such a study would be the only way of conclusively answering the question posed in the paper of "whether the introduction of genuine prior information into the inverse problem results in solutions that are more appealing than those found in other, heuristic regularization methods" (lines 92-94), similar to the study by Gudmundsson and Raymond (2008) to evaluate the use of such methods on uncovering basal topography and basal friction.

This being said, if the authors choose not to include something like this (which is ultimately up to the authors' discretion), I would recommend a reframing of the present study. As illustrated by the lines I cited above, the study is currently motivated by this question of determining whether including fractures as prior information can improve the inverse method. I'm not convinced that the methods the current manuscript uses actually answers this question (and, as a side note, I'm not sure I understand what "appealing" means in this context). I would perhaps recommend a slight reframing of the scientific question at hand to be less focused on testing the use of prior information and more focused on, as an example, evaluating different ice softness fields that can be recovered in Pine Island Glacier and what these different softness fields mean for the use of inverse methods to estimate ice softness. Of course, the exact framing I leave up to the authors, but I think it's important to make sure that the question posed is answered in the manuscript. I also think including the discussion of why the authors chose this methodology (e.g. Section 4.4: Next steps) should be included far earlier in the manuscript, if the authors choose not to include synthetic experiments, to illuminate for the reader why the methodology was chosen near where the methodology is described.

Thank you very much for this considered and well-reasoned assessment of the work. The work of Gudmundsson and Raymond (2008) provides a promising template for an alternative format for this kind of study, though it would change the article substantially to go down that route and would be, I think, unnecessary. The deficiencies pointed out in the current approach are, to some degree, addressed with that kind of experimental set up, though it replaces them with the difficulties in translating the intuition to the real world, as we have discussed before. As such, I think there is merit in the approach we take here and we are justified in leaving alternatives to future work. The new section "A synthetic example" at least shows that the idea makes sense where we have strong priors on where the ice is likely to be damaged, and also provides additional context to the approach we take throughout the rest of the article. With this, I am of the opinion that we do answer the question we pose in the article, all be it, broadly qualitatively.

As one final note, it may be worth it to contend in a bit more detail with the Gerli et al. 2024 study. I know the authors reference this in line 394, but an added sentence or two to explain why their results deviate from the results of the Gerli study would be enlightening. I leave this up to the authors' discretion, however, since this wasn't mentioned in my original review.

I have added and additional sentence to the discussion noting that the contradiction between the two studies shouldn't be too surprising given the degeneracy in solutions. I am hesitant to provide too strong a critique of that work despite thinking its conclusions are slightly unhelpful. I hope the general points about the importance of priors and the dangers of over-interpreting the solutions to ill-posed problems come across in the article.

**References**

Gudmundsson, G. H. and Raymond, M.: On the limit to resolution and information on basal properties obtainable from surface data on ice streams, The Cryosphere, 2, 167–178, https://doi.org/10.5194/tc-2-167-2008, 2008.